# OVERCOMING LABEL SHIFT IN TARGETED FEDERATED LEARNING

## ABSTRACT

Federated learning enables multiple actors to collaboratively train models without sharing private data. This unlocks the potential for scaling machine learning to diverse applications. Existing algorithms for this task are well-justified when clients and the intended target domain share the same distribution of features and labels, but this assumption is often violated in real-world scenarios. One common violation is label shift, where the label distributions differ across clients or between clients and the target domain, which can significantly degrade model performance. To address this problem, we propose FedPALS, a novel model aggregation scheme that adapts to label shifts by leveraging knowledge of the target label distribution at the central server. Our approach ensures unbiased updates under stochastic gradient descent, ensuring robust generalization across clients with diverse, label-shifted data. Extensive experiments on image classification demonstrate that Fed-PALS consistently outperforms standard baselines by aligning model aggregation with the target domain. Our findings reveal that conventional federated learning methods suffer severely in cases of extreme client sparsity, highlighting the critical need for target-aware aggregation. FedPALS offers a principled and practical solution to mitigate label distribution mismatch, ensuring models trained in federated settings can generalize effectively to label-shifted target domains.

## 1 INTRODUCTION

Federated learning has become a prominent paradigm in machine learning, enabling multiple clients to collaboratively train models without sharing their data (Kairouz et al., 2021). Central to this development is the widely-used *federated averaging* (FedAvg) algorithm, which aggregates model updates from clients, weighted by their data set sizes (McMahan et al., 2017). This aggregation rule is well-justified when client data is independent and identically distributed (i.i.d.) and has been effective in diverse domains such as healthcare (Sheller et al., 2020), finance risk prediction (Byrd & Polychroniadou, 2020), and natural language processing (Hilmkil et al., 2021).

The justification for FedAvg is weaker when clients exhibit systematic data heterogeneity, such as in the case of *label shift* (Zhao et al., 2018; Woodworth et al., 2020), since the learning objectives of clients differ from the objective optimized by the central server. Consider a federated learning task involving multiple retail stores (clients) where the goal is to predict the sales of different products based on customer purchase history, and deploy the trained model in a new store (target). Each store's sales reflect local preferences, leading to differences in label distributions and empirical risks. When label distributions vary substantially, the performance of FedAvg can be severely hampered (Karimireddy et al., 2020; Zhao et al., 2018). So, how can models be trained effectively across such heterogeneous clients? Several methods have been proposed to deal with client heterogeneity, such as regularization (Li et al., 2020a; 2021), clustering (Ghosh et al., 2020; Vardhan et al., 2024), and meta-learning approaches (Chen et al., 2018; Jiang et al., 2019; Fallah et al., 2020a). Still, these studies and prior work in federated learning assume that the target (or test) domain shares the same distribution as the combined training data from the clients. In many real-world applications, models must generalize to new target domains with different distributions and without training data.

In the retail example, the target store has no individual transaction data available and is not involved in training the model. The challenge of generalizing under distributional domain shift without target data is not exclusive to federated learning but arises also in centralized settings (Blanchard et al.,

2011; Muandet et al., 2013; Ganin et al., 2016). In centralized scenarios, domain adaptation techniques such as sample re-weighting (Lipton et al., 2018a), or domain-invariant representations (Arjovsky et al., 2020) can be employed to mitigate the effects of distributional shifts. However, these approaches require central access to observations from both the source and target domains. This is not feasible in federated learning due to the decentralized nature of the data. Neither the server nor the clients have access to both data sets, making direct application of these methods impractical.

**Contributions**   This work aims to improve the generalization of federated learning to target domains under *label shift*, in settings where the different label distributions of clients and target domains are known to the central server but unknown to the clients (see Section 2). To address this problem, we propose a novel aggregation scheme called FedPALS that optimizes a convex combination of client models to ensure that the aggregated model is better suited for the label distribution of the target domain (Section 3.1). Our approach is both well-justified and practical. We prove that the resulting stochastic gradient update behaves, in expectation, as centralized learning in the target domain (Proposition 1), and examine its relation to standard federated averaging (Proposition 3.2). We demonstrate the effectiveness of FedPALS through an extensive empirical evaluation (Section 5), showing that it outperforms traditional approaches in scenarios where distributional shifts pose significant challenges. Moreover, we observe that traditional methods struggle particularly in scenarios where the training clients have sparse label distributions. As we show in Figures 2c and 5c in Section 5, performance drops sharply when a majority of labels are completely unrepresented in clients, highlighting the limitations of existing approaches under label shift with extreme client sparsity.

## 2   TARGETED FEDERATED LEARNING UNDER LABEL SHIFT

Federated learning is a distributed machine learning paradigm wherein a global model $h_\theta$ is trained at a central server by aggregating parameter updates from multiple clients (McMahan et al., 2017). We focus on classification tasks in which the goal is for $h_\theta$ to predict the most probable label $Y \in \{1, ..., K\}$ for a given input $X \in \mathcal{X}$. Each client $i = 1, ..., M$ holds a data set $D_i = \{(x_{i,1}, y_{i,1}), ..., (x_{i,n_i}, y_{i,n_i})\}$ of $n_i$ labeled examples. Due to privacy concerns or limited communication bandwidth, these data sets cannot be shared with other clients or with the central server. Learning proceeds over rounds $t = 1, ..., t_{max}$, each completing three steps: (1) The central server broadcasts the current global model parameters $\theta_t$ to all clients; (2) Each client $i$ computes updated parameters $\theta_{i,t}$ based on their local data set $D_i$ and sends these updates back to the server; (3) The server aggregates the clients' updates to obtain the new global model $\theta_{t+1}$.

Client samples $D_i$ are assumed to be drawn i.i.d. from a *local client-specific* distribution $S_i(X, Y)$. A common (implicit) assumption in federated learning is that the learned model will be applied in a target domain $T(X, Y)$ that coincides with the marginal distribution of clients, $\bar{S} = \sum_{i=1}^{M} \frac{n_i}{N} S_i$, where $N = \sum_{i=1}^{M} n_i$. This is reflected in trained models being evaluated in terms of their average performance over all clients. In general, the target domain $T(X, Y)$ may be distinct from both individual clients and their aggregate (Bai et al., 2024). Moreover, clients may exhibit significant heterogeneity in their distributions, $S_i \neq S_j$ (Karimireddy et al., 2020; Li et al., 2020b). We refer to this problem as *targeted federated learning*.

**We study targeted federated learning under known label shift**, where no samples from the joint distribution $T(X, Y)$ are available but the target label distribution $T(Y)$ is known to the server. Let $\mathcal{X} \subset \mathbb{R}^d$ denote the $d$-dimensional input space and $\mathcal{Y} = \{1, ..., K\}$ the label space. Given a set of clients with distributions $S_1, ..., S_M$ and a target domain with distribution $T$, our objective is to minimize the expected risk, $R_T$ of a classifier $h_\theta : \mathcal{X} \to \mathcal{Y}$, with respect to a loss function $\ell : \mathcal{Y} \times \mathcal{Y} \to \mathbb{R}$ over the target distribution $T$,

$$\underset{\theta}{\text{minimize }} R_T(h_\theta) \coloneqq \underset{(X,Y) \sim T}{\mathbb{E}} [\ell(h_\theta(X), Y)] \ . \tag{1}$$

Obtaining a good estimate of $T(Y)$ is often feasible since it represents aggregate statistics that can be collected without the need for a large dataset. In our retail example, $Y = y$ would represent a sale in a specific product category, and $T(y)$ would correspond to the proportion of total sales in that category. A company could estimate these proportions without logging customer features $X$. Our setting differs from domain generalization which lacks a specific target domain (Bai et al., 2024).

In our setting, the target and all client label distributions are all assumed distinct: $\forall i \neq j \in [M]$ : $S_i(Y) \neq S_j(Y) \neq T(Y)$. Furthermore, the target distribution is assumed to differ from the client aggregate, $T(Y) \neq \bar{S}(Y)$. While the server has access to all marginal label distributions $S_i(Y)_{i=1}^M$ and $T(Y)$, these are *not available to the clients*. For instance, most retailers would be hesitant to share their exact sales statistics $T(Y)$ with competitors. Instead, they might provide this information to a neutral third party (central server) responsible for coordinating the federated learning process. More broadly, clients $i \neq j$ do not communicate directly with each other but rather interact with the central server through model parameters. While it is technically possible for the server to *infer* each client's label distribution $S_i(Y)$ based on their parameter updates (Ramakrishna & Dán, 2022), doing so would likely be considered a breach of trust in practical applications.

The distributional shifts between clients and the target are restricted to *label shift*—while the label distributions vary across clients and the target, the class-conditional input distributions are identical.

**Assumption 1** (Label shift). *For the client distributions $S_1, ..., S_M$ and the target distribution $T$,*

$$\forall i, j \in [M] : S_i(X \mid Y) = S_j(X \mid Y) = T(X \mid Y) . \tag{2}$$

This setting has been well studied in non-federated learning, both in cases where the label marginals are known and where they are not (Lipton et al., 2018b). In the retail example, label shift would mean that while the proportion of sales across product categories ($S_i(Y)$ and $T(Y)$) varies between different retailers and the target, the purchasing patterns within each category ($S_i(X \mid Y)$ and $T(X \mid Y)$) remain consistent. In other words, although retailers may sell different quantities of products across categories, the characteristics of customers buying a particular product (conditional on the product category) are assumed to be the same. Note that there are settings where both the label shift and *covariate shift* assumptions hold, that is $\forall i : S_i(Y \mid X) = T(Y \mid X)$ and $S_i(X \mid Y) = T(X \mid Y)$, but $S_i(X), T(X)$ differ, such as when the labeling function is deterministic. We do not consider general covariate shift (without the label shift assumption) here.

**Our central question is:** In federated learning, how can we *aggregate* the parameter updates $\theta_{i,t}$ of the $M$ clients, whose data sets are drawn from distributions $S_1, ..., S_M$, such that the resulting federated learning algorithm minimizes the target risk, $R_T$? The fact that clients are ignorant of the shift in $T(Y)$ affects the optimal strategy; direct access to $T(Y)$ would allow sample re-weighting or upsampling in the client objectives (Rubinstein & Kroese, 2016). This is not possible here.

## 3 FEDPALS: PARAMETER AGGREGATION TO ADJUST FOR LABEL SHIFT

In classical federated learning, all clients and the target domain are assumed i.i.d., and thus, the target risk (equation 1) is equal to the expected risk in any client

$$R_T(h) = \underset{(X,Y) \sim S_i}{\mathbb{E}} [\ell(h(X), Y)] =: R_i(h), \ \text{ for all } i = 1, ..., M .$$

Similarly, the empirical risk $\hat{R}_i$ of any client $i$ is identical in distribution (denoted $\overset{d}{=}$) to the empirical risk evaluated on a hypothetical data set $D_T = \{(x_{T,j}, y_{T,j})\}_{j=1}^{n_T}$ drawn from the target domain,

$$\hat{R}_i := \frac{1}{n_i} \sum_{j=1}^{n_i} \ell(h(x_{i,j}), y_{i,j}) \ \overset{d}{=} \ \frac{1}{n_T} \sum_{j=1}^{n_T} \ell(h(x_{T,j}), y_{T,j}) =: \hat{R}_T$$

As a result, if clients perform a single local gradient descent update, any convex combination of these gradients (updates) is equal in distribution to a classical (centralized) batch update for the target domain, given the previous parameter value,

$$\forall \alpha \in \Delta^{M-1} : \sum_i \alpha_i \nabla_\theta \hat{R}_i(h_\theta) \overset{d}{=} \nabla_\theta \hat{R}_T(h_\theta) ,$$

where $\Delta^{M-1} = \{\alpha \in [0,1]^M : \sum_i \alpha_i = 1\}$ is the simplex over $M$ elements. This property justifies the federated stochastic gradient (FedSGD) and FedAvg principles (McMahan et al., 2017),[1] which

---

[1]Strictly speaking, only FedSGD uses the property directly, but FedAvg is a natural extension.

aggregate model parameter updates through a convex combination, chosen to give weight to clients proportional to their sample size,

$$\theta_{t+1}^{FA} = \sum_{i=1}^{M} \alpha_i^{FA} \theta_{i,t} \quad \text{where} \quad \alpha_i^{FA} = \frac{n_i}{\sum_{j=1}^{M} n_j} \ . \tag{3}$$

A limitation of FedAvg aggregation is that when client and target domains are not identically distributed, and $T \neq \bar{S}$, $\{\nabla \hat{R}_i\}_{i=1}^{M}$ are no longer unbiased estimates of the risk gradient in the target domain, $\nabla R_T$. As a result, the FedAvg update is not an unbiased estimate of a model update computed in the target domain. As we see in Table 1 in section 5, this can have large effects on model quality.

### 3.1 OVERCOMING LABEL SHIFT WHILE CONTROLLING VARIANCE

Next, we develop an alternative aggregation strategy that partially overcomes the limitations of FedAvg under *label shift* (Assumption 1). Under label shift, it holds that

$$\forall i : R_T(h) = \sum_{y=1}^{K} T(y) \int_x \underbrace{T(x \mid y)}_{=S_i(x|y)} \ell(h(x), y) dx = \sum_{y=1}^{K} T(y) \mathbb{E}_{S_i}[\ell(h(X), y) \mid Y = y] \ .$$

In centralized (non-federated) learning under label shift, this insight is often used to re-weight (Lipton et al., 2018b) or re-sample (Japkowicz & Stephen, 2002) the empirical risk in the source domain. *This is not an option here since $T(Y)$ is not revealed to the clients.* For now, assume instead that the target label distribution can be covered by a convex combination of client label distributions. For this, let $\text{Conv}(S)$ denote the convex hull of distributions $\{S_i(Y)\}_{i=1}^{M}$.

**Assumption 2** (Target coverage). *For the label marginals $S_1(Y), ..., S_M(Y), T(Y) \in \Delta^{K-1}$, the target label distribution $T(Y)$ is covered by a convex combination $\alpha^c$ of client label distributions,*

$$T \in \text{Conv}(S), \quad i.e., \quad \exists \alpha^c \in \Delta^{M-1} : T(y) = \sum_{i=1}^{M} \alpha_i^c S_i(y) \quad \forall y \in [K] \ . \tag{4}$$

Note that, under label shift, Assumption 2 implies that $T(X, Y) = \sum_{i=1}^{M} \alpha_i^c S_i(X, Y)$, as well. Thus, under Assumptions 1–2, we have for any $\alpha^c$ satisfying equation 4,

$$R_T(h) = \sum_{y=1}^{K} \left( \sum_{i=1}^{M} \alpha_i^c S_i(y) \right) \mathbb{E}[\ell(h(X), y) \mid Y = y] = \sum_{i=1}^{M} \alpha_i^c R_{S_i}(h). \tag{5}$$

By extension, aggregating client updates with weights $\alpha^c$ will be an unbiased estimate of the update.

**Proposition 1** (Unbiased SGD update). *Consider a single round $t$ of federated learning in the batch stochastic gradient setting with learning rate $\eta$. Each client $i \in [M]$ is given parameters $\theta_t$ by the server, computes their local gradient, and returns the update $\theta_{i,t} = \theta_t - \eta \nabla_\theta \hat{R}_i(h_{\theta_t})$. Let weights $\alpha^c$ satisfy $T(X, Y) = \sum_{i=1}^{M} \alpha_i^c S_i(X, Y)$. Then, the aggregate update $\theta_{t+1} = \sum_{i=1}^{M} \alpha_i^c \theta_{i,t}$ satisfies*

$$\mathbb{E}[\theta_{t+1} \mid \theta_t] = \mathbb{E}[\theta_{t+1}^T \mid \theta_t] \ ,$$

*where $\theta_{t+1}^T = \theta_t - \eta \nabla_\theta \hat{R}_T(h_{\theta_t})$ is the batch stochastic gradient descent (SGD) update for $\hat{R}_T$ that would be obtained with a sample from the target domain. A proof is given in Appendix C.*

By Proposition 1, we are justified[2] in replacing the aggregation step of FedAvg with one where clients are weighted by $\alpha^c$. However, Assumption 2 may not hold, and $\alpha^c$ may not exist. For instance, if the target marginal is sparse, only clients with *the exact same sparsity pattern* as $T$ can be used in a convex combination $\alpha^c S = T$. That is, if we aim to classify images of animals and $T$ contains no tigers, then no clients contributing to the combination can have data containing tigers. At least, since $\{S_i(Y)\}_{i=1}^{M}, T(Y)$ are known to the server, it is straightforward to verify Assumption 2.

---

[2]Note that, just like in FedAvg, aggregating client parameter updates resulting from multiple local stochastic gradient steps (e.g., one epoch) is not guaranteed to locally minimize the target risk.

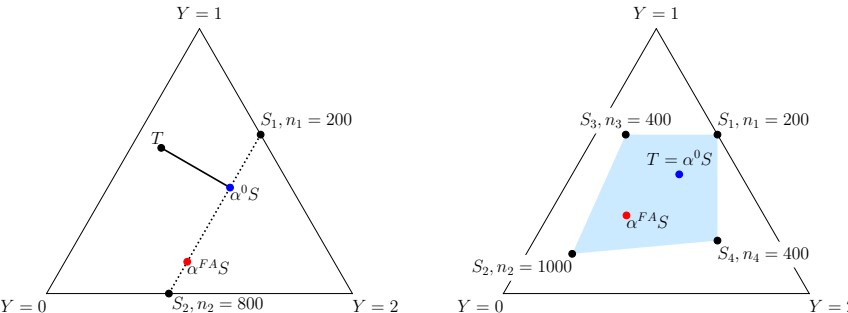

Figure 1: Illustration of the target label marginal $T$ and client marginals $S_1, ..., S_4$ in a ternary classification task, $Y \in \{0, 1, 2\}$. Left: there are fewer clients than labels, $M < K$, and $T \notin \text{Conv}(S)$; $\alpha^0 S$ is a projection of $T$ onto $\text{Conv}(S)$. Right: $T \in \text{Conv}(S)$ and coincides with $\alpha^0 S$. In both cases, the label marginal $\alpha^{FA} S$ implied by FedAvg is further from the target distribution.

**Remark.** In practice, many federated learning systems, including FedAvg, allow clients to perform several steps of local optimization (e.g., over an entire epoch) before aggregating the parameter updates at the server. This is a limitation of Proposition 1: when clients are not constrained to a single SGD step, the aggregated updates no longer strictly minimize the target risk.

A pragmatic choice when Assumption 2 is violated is to look for the convex combination $\alpha^0$ that most closely aligns with the target label distribution, and use that in the aggregation step,

$$\alpha^0 = \arg\min_{\alpha \in \Delta^{M-1}} \left\| \sum_{i=1}^{M} \alpha_i S_i(Y) - T(Y) \right\|_2^2 \quad \text{and} \quad \theta_{t+1}^0 = \sum_i \alpha_i^0 \theta_{i,t} . \tag{6}$$

We illustrate the label distributions implied by weighting with $\alpha^0$ and $\alpha^{FA}$ (FedAvg) in Figure 1.

**Effective sample size of aggregates.** A limitation of aggregating using $\alpha^0$ as defined in equation 6 is that, unlike FedAvg, it does not give higher weight to clients with larger sample sizes, which can lead to a higher variance in the model estimate. The variance of importance-weighted estimators can be quantified through the concept of *effective sample size* (ESS) (Kong, 1992), which measures the number of samples needed from the target domain to achieve the same variance as a weighted estimate computed from source-domain samples. ESS is often approximated as $1/(\sum_{j=1}^{m} w_j^2)$ where $w$ are normalized sample weights such that $w_j \geq 0$ and $\sum_{j=1}^{n} w_j = 1$. In federated learning, we can interpret the aggregation step as assigning a total weight $\alpha_i$ to each client $i$, which has $n_i$ samples. Consequently, each sample $(x_j, y_j) \in D_i$ has the same weight $\tilde{w}_j = \alpha_i/n_i$. The ESS for the aggregate is then given by $1/(\sum_{i=1}^{m} (\sum_{j \in S_i} \tilde{w}_j^2)) = 1/(\sum_{i=1}^{m} n_i \alpha_i^2/n_i^2) = 1/(\sum_{i=1}^{m} \alpha_i^2/n_i)$.

In light of the results above, we propose a client aggregation step such that the weighting of clients' label distributions will a) closely align with the target label distribution, and b) minimize the variance due to weighting using the inverse of the ESS. For a given regularization parameter $\lambda \in [0, \infty)$, we define a weighting scheme $\alpha^\lambda$ as the solution to the following problem.

$$\alpha^\lambda = \arg\min_{\alpha \in \Delta^{M-1}} \|T(Y) - \sum_{i=1}^{M} \alpha_i S_i(Y)\|_2^2 + \lambda \sum_i \frac{\alpha_i^2}{n_i} , \tag{7}$$

and aggregate client parameters inside federated learning as $\theta_{t+1}^\lambda = \sum_{i=1}^{M} \alpha_i^\lambda \theta_{i,t}$. We refer to this strategy as Federated learning with Parameter Aggregation for Label Shift (FedPALS).

### 3.2 FEDPALS IN THE LIMITS

In the FedPALS aggregation scheme (equation 7), there exists a trade-off between closely matching the target label distribution and minimizing the variance of the model parameters. This trade-off gives rise to two notable limit cases: $T \in \text{Conv}(S), \lambda \to 0$, and $\lambda \to \infty$. If all source distributions $\{S_i\}_{i=1}^{M}$ are identical and match the target distribution, this corresponds to the classical i.i.d. setting.

**Case 1:** $\lambda \to \infty \Rightarrow$ **Federated averaging**    In the limit $\lambda \to \infty$, as the regularization parameter $\lambda$ grows large, FedPALS aggregation approaches FedAvg aggregation.

**Proposition 2.** *The limit solution $\alpha^\lambda$ to equation 7, as $\lambda \to \infty$, is*

$$\lim_{\lambda \to \infty} \alpha_i^\lambda = \frac{n_i}{\sum_{j=1}^M n_j} = \alpha_i^{FA} \quad for \quad i = 1, \ldots, M . \tag{8}$$

The result is proven in Appendix C. By Proposition 2, the FedAvg weights $\alpha^{FA}$ minimize the ESS and coincide with FedPALS weights $\alpha^\lambda$ in the limit $\lambda \to \infty$. As a rare special case, whenever $T(Y) = \bar{S} = \sum_{i=1}^M \frac{n_i}{N} S_i(Y)$, FedAvg weights $\alpha^{FA} = \alpha^\lambda$ for any value of $\lambda$, since both terms attain their mimima at this point. However, this violates the assumption that $T(Y) \neq \bar{S}(Y)$.

**Case 2: Covered target,** $T \in \mathrm{Conv}(S)$    Now, consider when the target label distribution is in the convex hull of the source label distributions, $\mathrm{Conv}(S)$. Then, we can find a convex combination $\alpha^c$ of source distributions $S_i(Y)$ that recreate $T(Y)$, that is, $T(Y) = \sum_{i=1}^M \alpha_i^c S_i(Y)$. However, when there are more clients than labels, $M > K$, such a *satisfying combination* $\alpha^c$ need not be unique and different combinations may have different effective sample size. Let $A^c = \{\alpha^c \in \Delta^{M-1} : T(Y) = (\alpha^c)^\top S(Y)\}$ denote all satisfying combinations where $S(Y) \in \mathbb{R}^{M \times K}$ is the matrix of all client label marginals. For a sufficiently small regularization penalty $\lambda$, the solution to equation 7 will be the satisfying combination with largest effective sample size.

$$\lim_{\lambda \to 0} \alpha^\lambda = \arg\min_{\alpha \in A^c} \sum_{i=1}^M \frac{\alpha_i^2}{n_i} .$$

If there are fewer clients than labels, $M < K$, the set of target distributions for which a satisfying combination exists has measure zero, see Figure 1 (left). Nevertheless, the two cases above allow us to interpolate between being as faithful as possible to the target label distribution ($\lambda \to 0$) and retaining the largest effective sample size ($\lambda \to \infty$), the latter coinciding with FedAvg.

Finally, when $T \in \mathrm{Conv}(S)$ and $\lambda \to 0$, Proposition 1 applies also to FedPALS; the aggregation strategy results in an unbiased estimate of the target risk gradient in the SGD setting. However, like the unregularized weights, Proposition 1 does not apply for multiple local client updates.

**Case 3:** $T \notin \mathrm{Conv}(S)$    If the target distribution does not lie in $\mathrm{Conv}(S)$, see Figure 1 (left), FedPALS projects the target to the "closest point" in $\mathrm{Conv}(S)$ if $\lambda = 0$, and to a tradeoff between this projection and the FedAvg aggregation if $\lambda > 0$.

**Choice of hyperparameter** $\lambda$    A salient question in Cases 2 & 3 is how to choose the strength of the regularization, $\lambda$. A larger value will generally favor influence from more clients, provided that they have sufficiently many samples. When $T \notin \mathrm{Conv}(S)$, the convex combination closest to $T$ could have weight on a single vertex. This will likely hurt the generalizability of the resulting classifier. In experiments, we compare values of $\lambda$ that yield different effective sample sizes, such as 10%, 25%, 50%, 75%, or 100% of the original sample size, $N$. We can find these using binary search by solving equation 7 and calculate the ESS. One could select $\lambda$ heuristically based on the the ESS, or treat $\lambda$ as a hyperparameter and select it using a validation set. Although this would entail training and evaluating several models which can be seen as a limitation. We elect to choose a small set of $\lambda$ values based on the ESS heuristic and train models for these. Then we use a validation set to select the best performing model. This highlights the usefulness of the ESS as a heuristic. If it is unclear which values to pick, one could elect for a simple strategy of taking the ESS of $\lambda = 0$ and 100% and taking $l$ equidistributed values in between the two extremes, for some small integer $l$.

**Sparse clients and targets**    In problems with a large number of labels, $K \gg 1$, it is common that any individual domain (clients or target) supports only a subset of the labels. For example, in the IWildCam benchmark, not every wildlife camera captures images of all animal species. When the target $T(Y)$ is *sparse*, meaning $T(y) = 0$ for certain labels $y$, it becomes easier to find a good match $(\alpha^\lambda)^\top S(Y) \approx T(Y)$ if the client label distributions are also sparse. Achieving a perfect match, i.e., $T \in \mathrm{Conv}(S)$, requires that (i) the clients collectively cover all labels in the target, and (ii) each client contains only labels that are present in the target. If this is also beneficial for learning, it would suggest that the client-presence of labels that are not present in the target would *harm* the aggregated model. We study the implications of sparsity of label distributions empirically in Section 5.

## 4 RELATED WORK

Efforts to mitigate the effects of distributional shifts in federated learning can be broadly categorized into client-side and server-side approaches. Client-side methods use techniques such as clustering clients with similar data distributions and training separate models for each cluster (Ghosh et al., 2020; Sattler et al., 2020; Vardhan et al., 2024), and meta-learning to enable models to quickly adapt to new data distributions with minimal updates (Chen et al., 2018; Jiang et al., 2019; Fallah et al., 2020b). Other notable strategies include logit calibration (Zhang et al., 2022), regularization techniques that penalize large deviations in client updates to ensure stable convergence (Li et al., 2020b; 2021), and recent work on optimizing for flatter minima to enhance model robustness (Qu et al., 2022; Caldarola et al., 2022). Server-side methods focus on improving model aggregation or applying post-aggregation adjustments. These include optimizing aggregation weights (Reddi et al., 2021), learning adaptive weights (Li et al., 2023), iterative moving averages to refine the global model (Zhou et al., 2023), and promoting gradient diversity during updates (Zeng et al., 2023). Both categories of work overlook shifts in the target distribution, leaving this area unexplored.

Another related area is personalized federated learning, which focuses on fine-tuning models to optimize performance on each client's specific local data (Collins et al., 2022; Boroujeni et al., 2024; Fallah et al., 2020a). This setting differs fundamentally from our work, which focuses on improving generalization to new target clients without any training data available for fine-tuning. Label distribution shifts have also been explored with methods such as logit calibration (Zhang et al., 2022; Wang et al., 2023; Xu et al., 2023), novel loss functions (Wang et al., 2021), feature augmentation (Xia et al., 2023), gradient reweighting (Xiao et al., 2023), and contrastive learning (Wu et al., 2023). However, like methods aimed at mitigating the effects of general shifts, these do not address the challenge of aligning models with an unseen target distribution, as required in our setting.

Generalization under domain shift in federated learning remains underdeveloped (Bai et al., 2024). The work most similar to ours is that of agnostic federated learning (AFL) (Mohri et al., 2019), which aims to learn a model that performs robustly across all possible target distributions within the convex hull of client distributions. One notable approach is tailored for medical image segmentation, where clients share data in the frequency domain to achieve better generalization across domains (Liu et al., 2021). However, this technique requires data sharing, making it unsuitable for privacy-sensitive applications like ours. A different line of work focuses on addressing covariate shift in federated learning through importance weighting (Ramezani-Kebrya et al., 2023). Although effective, this method requires sending samples from the test distribution to the server, which violates our privacy constraints.

## 5 EXPERIMENTS

We perform a series of experiments on benchmark data sets to evaluate FedPALS in comparison with baseline federated learning algorithms. The experiments aim to demonstrate the value of the central server knowing the label distributions of the client and target domains when these differ substantially. Additionally, we seek to understand how the parameter $\lambda$, controlling the trade-off between bias and variance in the FedPALS aggregation scheme, impacts the results. Finally, we investigate how the benefits of FedPALS are affected by the sparsity of label distributions and by the distance $d(T, S) \coloneqq \min_{\alpha \in \Delta^{M-1}} \|T(Y) - \alpha^\top S(Y)\|_2^2$ from the target to the convex hull of clients.

**Experimental setup** While numerous benchmarks exist for federated learning (Caldas et al., 2018; Ogier du Terrail et al., 2022; Chen et al., 2022) and domain generalization (Gulrajani & Lopez-Paz, 2020; Koh et al., 2021), respectively, until recently none have addressed tasks that combine both settings. To fill this gap, Bai et al. (2024) introduced a benchmark specifically designed for federated domain generalization (DG), evaluating methods across diverse datasets with varying levels of client heterogeneity. In our experiments, we use the PACS Li et al. (2017) and iWildCAM data sets from the Bai et al. (2024) benchmark to model realistic label shifts between the client and target distributions. We modify the PACS dataset to consist of three clients, each missing a label that is present in the other two. Additionally, one client is reduced to one-tenth the size of the others, and the target distribution is made sparse in the same label as that of the smaller client. Further details are given in Appendix A.

Table 1: Comparison of mean accuracy and standard deviation ($\pm$) across different algorithms. The reported values are over 8 independent random seeds for the CIFAR-10 and Fashion-MNIST tasks, and 3 for PACS. $C$ indicates the number of labels per client and $\beta$ the Dirichlet concentration parameter. $M$ is the number of clients. The *Oracle* method refers to a FedAvg model trained on clients whose distributions are identical to the target.

| Data set | Label split | M | FedPALS | FedAvg | FedProx | SCAFFOLD | AFL | Oracle |
|---|---|---|---|---|---|---|---|---|
| Fashion-MNIST | $C = 3$ | 10 | $92.4 \pm 2.1$ | $67.1 \pm 22.0$ | $66.9 \pm 20.8$ | $69.5 \pm 19.3$ | $72.2 \pm 16.5$ | $97.6 \pm 2.1$ |
| | $C = 2$ | | $80.6 \pm 23.7$ | $53.9 \pm 36.2$ | $52.9 \pm 35.7$ | $54.9 \pm 36.8$ | $72.8 \pm 21.7$ | $97.5 \pm 4.0$ |
| CIFAR-10 | $C = 3$ | | $65.6 \pm 10.1$ | $44.0 \pm 8.4$ | $43.5 \pm 7.2$ | $43.3 \pm 7.4$ | $53.2 \pm 0.9$ | $85.5 \pm 5.0$ |
| | $C = 2$ | 10 | $72.8 \pm 17.4$ | $46.7 \pm 15.8$ | $47.7 \pm 15.6$ | $46.7 \pm 14.9$ | $54.7 \pm 0.1$ | $89.2 \pm 3.9$ |
| | $\beta = 0.1$ | | $62.6 \pm 17.9$ | $40.8 \pm 9.2$ | $41.9 \pm 9.7$ | $43.5 \pm 10.5$ | $53.4 \pm 11.5$ | $79.2 \pm 3.7$ |
| PACS | $C = 6$ | 3 | $86.0 \pm 2.9$ | $73.4 \pm 1.6$ | $75.3 \pm 1.3$ | $73.9 \pm 0.3$ | $74.5 \pm 0.9$ | $90.5 \pm 0.3$ |

Furthermore, we construct two additional tasks by introducing label shift to standard image classification data sets, Fashion-MNIST (Xiao et al., 2017) and CIFAR-10 (Krizhevsky, 2009). We apply two label shift sampling strategies: sparsity sampling and Dirichlet sampling. Sparsity sampling involves randomly removing a subset of labels from clients and the target domain, following the data set partitioning technique first introduced in McMahan et al. (2017) and extensively used in subsequent studies (Geyer et al., 2017; Li et al., 2020a; 2022). Each client is assigned $C$ random labels, with an equal number of samples for each label and no overlap among clients.

Dirichlet sampling simulates realistic non-i.i.d. label distributions by, for each client $i$, drawing a sample $p_i \sim \text{Dirichlet}_K(\beta)$, where $p_i(k)$ represents the proportion of samples in client $i$ that have label $k \in [K]$. We use a symmetric concentration parameter $\beta > 0$ which controls the sparsity of the client distributions. A smaller $\beta$ results in more heterogeneous client data sets, while a larger value approximates an i.i.d. setting. This widely-used method for sampling clients was first introduced by Yurochkin et al. (2019). While prior works have focused on inter-client distribution shifts assuming that client and target domains are equally distributed, *we apply these sampling strategies also to the target set*, thereby introducing label shift between the client and target data. Figures 2b & 5b (latter in appendix) illustrate an example with $C = 6$ for sparsity sampling and Dirichlet sampling with $\beta = 0.1$, where the last client (Client 9) is chosen as the target. In addition, we investigate the effect of $T(Y) \notin \text{Conv}(S)$ in a synthetic task described in B.4.

**Baseline algorithms and model architectures** Alongside FedAvg, we use SCAFFOLD, FedProx and AFL (Karimireddy et al., 2020; Li et al., 2020b; Mohri et al., 2019) as baselines, the first two chosen due to their prominence in the literature and AFL as it is similar in nature to FedPALS. SCAFFOLD mitigates client drift in heterogeneous data environments by introducing control variates to correct local updates. FedProx incorporates a proximal term to the objective to limit the divergence of local models from the global model. AFL optimizes the global model to perform well on an unknown target which is a combination of the clients. For the synthetic experiment in Section B.4, we use a logistic regression model. For CIFAR-10 and Fashion-MNIST, we use small, two-layer convolutional networks, while for PACS and iWildCAM, we use a ResNet-50 pre-trained on ImageNet. Early stopping, model hyperparameters, and $\lambda$ in FedPALS are tuned using a validation set that reflects the target distribution in the synthetic experiment, CIFAR-10, Fashion-MNIST, and PACS. This tuning process consistently resulted in setting the number of local epochs to $E = 1$ across all experiments. For iWildCAM, we adopt the hyperparameters reported by Bai et al. (2024) and select $\lambda$ using the same validation set used in their work. We report the mean test accuracy and standard deviation for each method over 3 independent random seeds for PACS and iWildCam and 8 seeds for the smaller Fashion-MNIST and CIFAR-10, to ensure a robust evaluation.

## 5.1 Experimental results on benchmark tasks

We present summary results for three tasks with selected skews in Table 1 and explore detailed results below. Across these tasks, FedPALS consistently outperforms or matches the best-performing baseline. For PACS, Fashion-MNIST and CIFAR-10, we include results for an *Oracle* FedAvg model, which is trained on clients whose distributions are identical to the target distribution, eliminating any client-target distribution shift (see Appendix A for details on its construction). A FedPALS *Oracle* would be equivalent since there is no label shift. The *Oracle*, which enjoys perfect

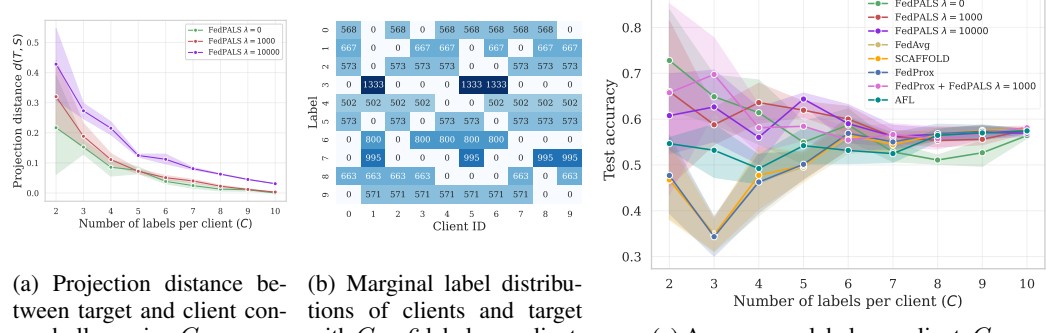

(a) Projection distance between target and client convex hull, varying $C$.

(b) Marginal label distributions of clients and target with $C = 6$ labels per client.

(c) Accuracy vs labels per client, $C$.

Figure 2: Results on CIFAR-10 with sparsity sampling, varying the number of labels per clients $C$ across 10 clients. Clients with IDs 0–8 are used in training, and Client 9 is the target client. The task is more difficult for small $C$, when fewer clients share labels, and the projection distance is larger.

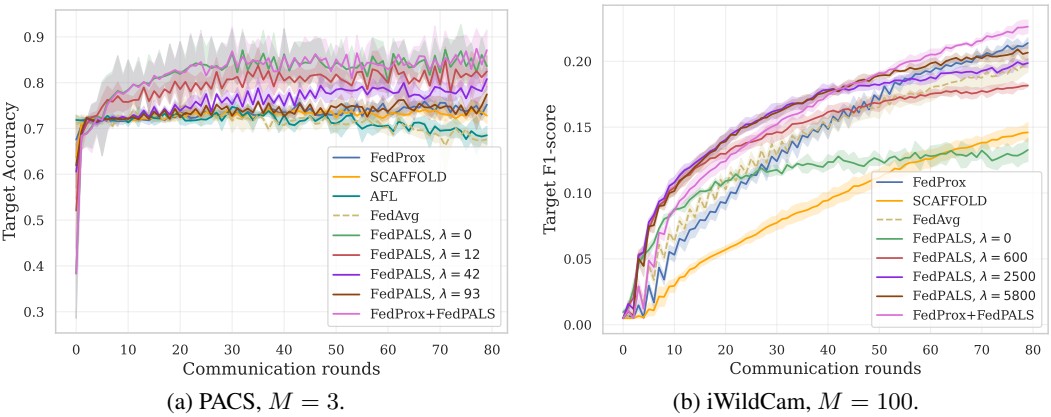

(a) PACS, $M = 3$.

(b) iWildCam, $M = 100$.

Figure 3: Target accuracy during training of FedPALS compared to baselines on PACS (a) and iWildCam (b), averaged over 3 random seeds. $M$ is the number of training clients.

alignment between client and target distributions, achieves superior performance, underscoring the challenges posed by distribution shifts in real-world scenarios where such alignment is absent.

**CIFAR-10/Fashion-MNIST.** Figure 2c shows the results for the CIFAR-10 data set, where we vary the label sparsity across clients. In the standard i.i.d. setting, where all labels are present in both the training and target clients ($C = 10$), all methods perform comparably. However, as label sparsity increases and fewer labels are available in client data sets (i.e., as $C$ decreases), we observe a performance degradation in standard baselines. In contrast, our proposed method, FedPALS, leverages optimized aggregation to achieve a lower target risk, resulting in improved test accuracy under these challenging conditions. Similar trends are observed for Fashion-MNIST, as shown in Figure 6 in Appendix B. Furthermore, the results in the highly non-i.i.d. cases ($C = 2, 3$ and $\beta = 0.1$) are summarized in Table 1. Additional experiments in Appendix B examine how the algorithms perform with varying numbers of local epochs and clients.

**PACS.** As shown in Figure 3a, being faithful to the target distribution is crucial for improved performance. Lower values of $\lambda$ generally correspond to better performance. Notably, FedAvg struggles in this setting because it systematically underweights the client with the distribution most similar to the target, leading to suboptimal model performance. In fact, this even causes performance to degrade over time. Interestingly, the baselines also face challenges on this task: both FedProx and SCAFFOLD perform similarly to FedPALS when $\lambda = 93$. However, FedPALS demonstrates significant improvements over these methods, highlighting the effectiveness of our aggregation scheme in enhancing performance. We also see that FedPALS + FedProx performs comparably to just using FedPALS in this case, although it does have higher variance. Additionally, in Table 1, we present the models selected based on the source validation set, where FedPALS outperforms all other methods.

For comprehensive results, including all FedPALS models and baseline comparisons, please refer to Table 3 in Appendix B.

**iWildCam.** The test performance across communication rounds is shown in Figure 3b. Initially, FedPALS widens the performance gap compared to FedAvg, but as training progresses, this gain diminishes. While FedPALS quickly reaches a strong performing model, it eventually plateaus. The rate of convergence and level of performance reached appears to be influenced by the choice of $\lambda$, with lower values of $\lambda$ leading to faster plateaus at lower levels compared to larger ones. This suggests that more uniform client weights and a larger effective sample size are preferable in this task. Given the iWildCam dataset's significant class imbalance – with many classes having few samples – de-emphasizing certain clients can degrade performance. We also note that our assumption of label shift need not hold in this experiment, as the cameras are in different locations, potentially leading to variations in the conditional distribution $p(X \mid Y)$. The performance of the models selected using the source validation set is shown in Table 2 in Appendix B. There we see that FedPALS performs comparably to FedAvg and FedProx while outperforming SCAFFOLD. Unlike in other tasks, where FedProx performs comparably or worse than FedPALS, we see FedProx achieve the highest F1-score on this task. Therefore, we conduct an additional experiment where we use both FedProx and FedPALS together, as they are not mutually exclusive. This results in the best performing model, see Figure 3b. Finally, as an illustration of the impact of increasing $\lambda$, we provide the weights of the clients in this experiment alongside the FedAvg weights in 4 in Appendix B. We note that as $\lambda$ increases, the weights increasingly align with those of FedAvg while retaining weight on the clients whose label distributions most resemble that of the target.

# 6 DISCUSSION

We explored *targeted federated learning under label shift*, a scenario where client data distributions differ from a target domain with a known label distribution, but no target samples are available. We demonstrated that traditional approaches, such as FedAvg, which assume identical distributions between clients and the target, fail to adapt effectively in this context due to biased aggregation of client updates. To address this, we proposed FedPALS, a novel aggregation strategy that optimally combines client updates to align with the target distribution, ensuring that the aggregated model minimizes target risk. Empirically, across diverse tasks, we showed that under label shift, FedPALS significantly outperforms standard methods like FedAvg, FedProx and SCAFFOLD, as well as AFL. Specifically, when the target label distribution lies within the convex hull of the client distributions, FedPALS finds the solution with the largest effective sample size, leading to a model that is most faithful to the target distribution. More generally, FedPALS balances the trade-off between matching the target label distribution and minimizing variance in the model updates. Our experiments further highlight that FedPALS excels in challenging scenarios where label sparsity and client heterogeneity hinder the performance of conventional federated learning methods.

We also observed that the choice of the trade-off parameter $\lambda$ is crucial for achieving optimal performance in tasks such as iWildCam, where the label shift assumption may not fully hold. Moreover, FedPALS can underperform in scenarios where one or more clients, which are essential for accurately mirroring the target distribution, have limited sample sizes, and $\lambda$ is set too low. In such cases, the effective sample size of the aggregated dataset becomes insufficient, potentially hindering the model's ability to learn effectively. Further, when the client aggregate is identical to the target, we do not expect this method to produce better solutions than FedAvg as the methods are equivalent in this case. Similar to many methods in FL there is an inherent privacy-accuracy trade-off where we achieve increased accuracy. However, this comes at the cost of clients sharing their label marginals.

Interestingly, the early performance gains observed during training suggest that dynamically tuning $\lambda$ over time could enhance performance of FedPALS. A promising avenue for future work would be exploring adaptive strategies for dynamically tuning $\lambda$. We also observed empirically that FedAvg tends to do much better when there is at least a small sample of each label in the clients making the gain from using FedPALS smaller. This could be further investigated to see if this behaviour can be replicated for more difficult tasks. Additionally, our weighting approach could be extended to the covariate shift setting, where input distributions vary between clients and the target. This extension is feasible when the central server has access to unlabeled target and client samples, akin to unsupervised domain adaptation in the centralized learning paradigm.

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

APPENDIX

# A  EXPERIMENTAL DETAILS

Here we provide additional details about the experimental setup for the different tasks.

## A.1  ORACLE CONSTRUCTION

The *Oracle* method serves as a benchmark to illustrate the performance upper bound when there is no distribution shift between the clients and the target. To construct this *Oracle*, we assume that the client label distributions are identical to the target label distribution, effectively eliminating the label shift that exists in real-world scenarios.

In practice, this means that for each dataset, the client data is drawn directly from the same distribution as the target. The aggregation process in the *Oracle* method uses FedAvg, as no adjustments for label shift are needed. Since the client and target distributions are aligned, FedPALS would behave equivalently to FedAvg under this setting, as there is no need for reweighting the client updates.

This method allows us to assess the maximum possible performance that could be achieved if the distributional differences between clients and the target did not exist. By comparing the *Oracle* results to those of our proposed method and other baselines, we can highlight the impact of label shift on model performance and validate the improvements brought by FedPALS.

## A.2  SYNTHETIC TASK

We randomly sampled three means $\mu_1 = [6, 4.6], \mu_2 = [1.2, -1.6]$, and $\mu_3 = [4.6, -5.4]$ for each label cluster, respectively.

## A.3  PACS

In this task we use the official source and target splits which are given in the work by Bai et al. (2024). We construct the task such that the training data is randomly assigned among three clients, then we remove the samples of one label from each of the clients. This is chosen to be labels '0', '1' and '2'. Then the client that is missing the label '2' is reduced so that it is $10\%$ the amount of the original size. For the target we modify the given one by removing the samples with label '2', thereby making it more similar to the smaller client. To more accurately reflect the target distribution we modify the source domain validation set to also lack the samples with label '2'. This is reasonable since we assume that we have access to the target label distribution.

We pick four values of $\lambda$, [0,12,42,93], which approximately correspond to an ESS of $15\%, 25\%, 50\%$ and $75\%$ respectively. We use the same hyperparameters during training as Bai et al. (2024) report using in their paper. Furthermore, we use the cross entropy loss in this task.

## A.4  iWILDCAM

We perform this experiment using the methodology described in Bai et al. (2024) with the heterogeneity set to the maximum setting, i.e., $\lambda = 0$ in their construction.[3] We use the same hyperparameters which is used for FedAvg in the same work to train FedPALS. We perform 80 rounds of training and, we then select the best performing model based on held out validation performance and report the mean and standard deviation over three random seeds. This can be seen in Table 2. We pick four values of $\lambda$, [0,600,2500,5800], which approximately correspond to an ESS of $8\%, 25\%, 50\%$ and $75\%$ respectively. We use the cross entropy loss in this task.

Due to FedProx performing comparably to FedPALS on this task, in contrast with other experiments, we also perform an experiment where we do both FedProx and FedPALS. This is easily done as FedProx is a client side method while FedPALS is a weighting method applied at the server. This results in the best performing model.

---

[3]Note that this is not the same $\lambda$ used in the trade-off in FedPALS.

We use the same hyperparameters during training as Bai et al. (2024) report using in their paper. However, we set the amount of communication rounds to 80.

# B ADDITIONAL EMPIRICAL RESULTS

Figure 4 illustrates the aggregation weights of clients in the iWildCam experiment for $\lambda$ corresponding to different effective sample sizes.

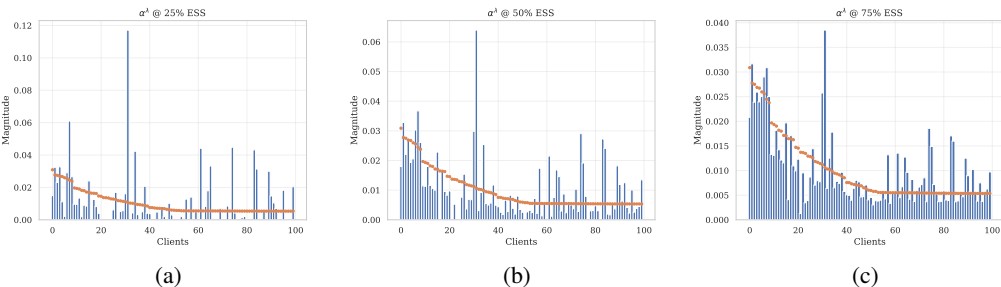

(a) (b) (c)

Figure 4: An illustration of the aggregation weights of clients in the iWildCam experiment using FedPALS for different ESS. The clients are sorted by amount of samples in descending order. The magnitude of the weights produced by federated averaging is shown as dots. Note that with increasing the ESS, the magnitudes more closely resemble that of federated averaging.

We report the performance of the models selected using the held out validation set in Table 2 and Table 3 for the iWildCam and PACS experiments respectively.

Table 2: Results on iWildCam with 100 clients, standard deviation reported over 3 random seeds.

| Algorithm | F1 (macro) |
|---|---|
| **FedPALS,** $\lambda = 0$ | $0.13 \pm 0.00$ |
| **FedPALS,** $\lambda = 600$ | $0.18 \pm 0.00$ |
| **FedPALS,** $\lambda = 2500$ | $0.19 \pm 0.00$ |
| **FedPALS,** $\lambda = 5800$ | $0.21 \pm 0.00$ |
| **FedProx+FedPALS,** $\lambda = 5800$ | $0.23 \pm 0.00$ |
| **FedAvg** | $0.20 \pm 0.01$ |
| **FedProx** | $0.21 \pm 0.00$ |
| **SCAFFOLD** | $0.15 \pm 0.01$ |

Table 3: Results on PACS with 3 clients with mean and standard deviation reported over 3 random seeds.

| Algorithm | Accuracy |
|---|---|
| **FedPALS,** $\lambda = 0$ | $86.0 \pm 2.9$ |
| **FedPALS,** $\lambda = 12$ | $84.3 \pm 2.5$ |
| **FedPALS,** $\lambda = 42$ | $81.7 \pm 1.2$ |
| **FedPALS,** $\lambda = 93$ | $77.3 \pm 1.6$ |
| **FedProx+FedPALS,** $\lambda = 0$ | $86.1 \pm 4.7$ |
| **FedAvg** | $73.4 \pm 1.6$ |
| **FedProx** | $75.3 \pm 1.3$ |
| **SCAFFOLD** | $73.9 \pm 0.3$ |
| **AFL** | $74.5 \pm 0.9$ |

## B.1 RESULTS ON CIFAR-10 WITH DIRICHLET SAMPLING

Figure 5 shows the results for the CIFAR-10 experiment with Dirichlet sampling of client and target label distributions.

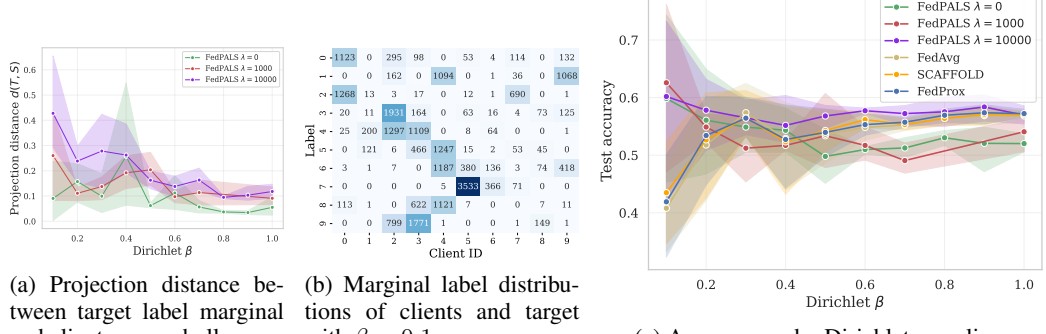

(a) Projection distance between target label marginal and client convex hull

(b) Marginal label distributions of clients and target with $\beta = 0.1$.

(c) Accuracy under Dirichlet sampling.

Figure 5: Results on CIFAR-10 with Dirichlet sampling across 10 clients, varying concentration parameter $\beta$. Clients with IDs 0–8 are clients present during training, and client with ID 9 is the target client.

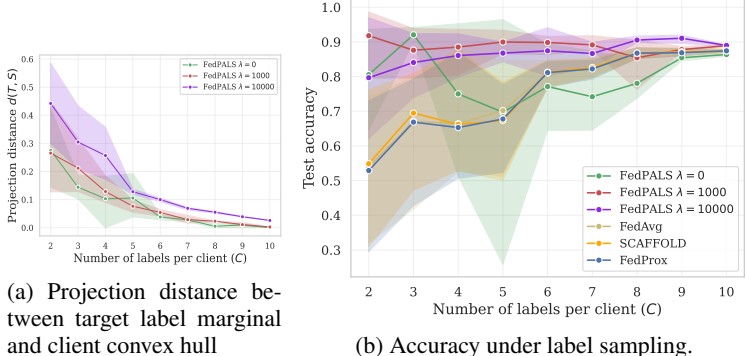

(a) Projection distance between target label marginal and client convex hull

(b) Accuracy under label sampling.

Figure 6: Results on Fashion-MNIST with label sampling across 10 clients, varying parameter $C$. Clients with IDs 0–8 are clients present during training, and client with ID 9 is the target client.

## B.2 TRAINING DYNAMICS FOR FASHION-MNIST

Figure 7 shows the training dynamics for Fashion-MNIST and CIFAR-10 with different label marginal mechanisms.

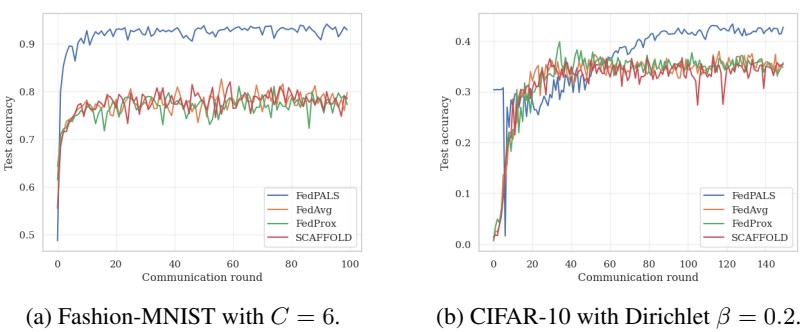

(a) Fashion-MNIST with $C = 6$.

(b) CIFAR-10 with Dirichlet $\beta = 0.2$.

Figure 7: Test accuracy during training rounds.

### B.3 LOCAL EPOCHS AND NUMBER OF CLIENTS

In Figure 8c we show results for varying number of clients for each method. For the cases with number of clients 50 and 100, we use the standard sampling method of federated learning where a fraction of 0.1 clients are sampled in each communication round. In this case, we optimize $\alpha^\lambda$ for the participating clients in each communication round. Interestingly, we observe that while FedAvg performs significantly worse than FedPALS on a target client under label shift, it outperforms both FedProx and SCAFFOLD when the number of local epochs is high ($E = 40$), as shown in Figure 8b.

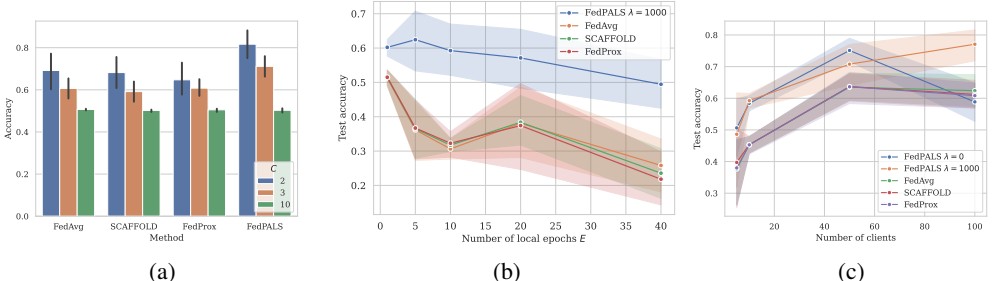

| (a) | (b) | (c) |

Figure 8: Comparison of CIFAR-10 results with different clients and settings. (a) 100 clients for $C = 2, 3, 10$, $\lambda = 1000$. (b) 10 clients and number of labels $C = 3$. We plot test accuracy as a function of number of local epochs $E$. The total number of communication rounds $T$ are set such that $T = E/150$, where 150 is the number of rounds used for $E = 1$. (c) Test accuracy as a function of number of clients, with $C = 3$.

### B.4 SYNTHETIC EXPERIMENT: EFFECT OF PROJECTION DISTANCE ON TEST ERROR

When the target distribution $T(Y)$ is not covered by the clients, FedPALS finds aggregation weights corresponding to a regularized projection of $T$ onto $\mathrm{Conv}(S)$. To study the impact of this, we designed a controlled experiment where the distance of the projection is varied. We create a classification task with three classes, $\mathcal{Y} = \{0, 1, 2\}$, and define $p(X \mid Y = y)$ for each label $y \in \mathcal{Y}$ by a unit-variance Gaussian distribution $\mathcal{N}(\mu_y, I)$, with randomly sampled means $\mu_y \in \mathbb{R}^2$. We simulate two clients with label distributions $S_1(Y) = [0.5, 0.5, 0.0]^\top$ and $S_2(Y) = [0.5, 0.0, 0.5]^\top$, and $n_1 = 40$, $n_2 = 18$ samples, respectively. Thus, FedAvg gives larger weight to Client 1. We define a target label distribution $T(Y)$ parameterized by $\delta \in [0, 1]$ which controls the projection distance $d(T, S)$ between $T(Y)$ and $\mathrm{Conv}(S)$,

$$T_\delta(Y) \coloneqq (1 - \delta)T_{\mathrm{proj}}(Y) + \delta T_{\mathrm{ext}}(Y) \,,$$

with $T_{\mathrm{ext}}(Y) = [0, 0.5, 0.5]^\top \notin \mathrm{Conv}(S(Y))$ and $T_{\mathrm{proj}}(Y) = [0.5, 0.25, 0.25]^\top \in \mathrm{Conv}(S(Y))$. By varying $\delta$, we control the projection distance $d(T, S)$ between each $T_\delta$ and $\mathrm{Conv}(S)$ from solving equation 6, allowing us to study its effect on model performance.

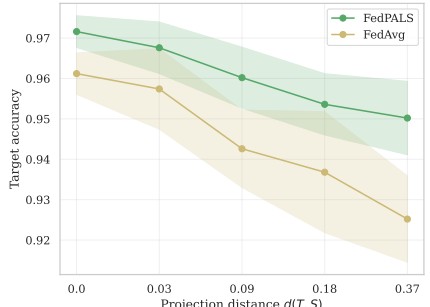

Figure 9: Synthetic experiment. Accuracy of the global model as a function of the projection distance $d(T, S)$ between the target distribution $T(Y)$ and client label distributions $\mathrm{Conv}(S(Y))$. Means and standard deviations reported over 5 independent runs.

We evaluate the global model on a test set with $n_{\mathrm{test}} = 2000$ samples drawn from the target distribution $T(Y)$ for each value of $\delta$ and record the target accuracy for FedPALS and FedAvg. Figure 9 illustrates the relationship between the target accuracy and the projection distance $d(T, S)$ due to varying $\delta$. When $d(S, T) = 0$ (i.e., $T(Y) \in \mathrm{Conv}(S)$), the target accuracy is highest, indicating that our method successfully matches the target distribution. As $d(S, T)$ increases (i.e., $T$ moves further away from $\mathrm{Conv}(S)$), the task becomes harder and accuracy declines. For all values, FedPALS performs better than FedAvg. For more details on the synthetic experiment, see Appendix A.

# C PROOFS

## C.1 FEDPALS UPDATES

**Proposition 1 (Repeated)** (Unbiased SGD update). Consider a single round $t$ of federated learning in the batch stochastic gradient setting with learning rate $\eta$. Each client $i \in [M]$ is given parameters $\theta_t$ by the server, computes their local gradient, and returns the update $\theta_{i,t} = \theta_t - \eta \nabla_\theta \hat{R}_i(h_{\theta_t})$. Let weights $\alpha^c$ satisfy $T(X,Y) = \sum_{i=1}^M \alpha_i^c S_i(X,Y)$. Then, the aggregate update $\theta_{t+1} = \sum_{i=1}^M \alpha_i^c \theta_{i,t}$ satisfies

$$\mathbb{E}[\theta_{t+1} \mid \theta_t] = \mathbb{E}[\theta_{t+1}^T \mid \theta_t],$$

where $\theta_{t+1}^T$ is the batch stochastic gradient update for $\hat{R}_T$ that would be obtained with a sample from the target domain.

*Proof.*

$$\theta_{t+1} = \sum_{i=1}^M \alpha_i^c \theta_{i,t} = \sum_{i=1}^M \theta_i^c(\theta_t - \eta \nabla \hat{R}_i(h_{\theta_t})) = \theta_t - \eta \sum_{i=1}^M \alpha_i \nabla \hat{R}_i(h_{\theta_t}) \tag{9}$$

$$\mathbb{E}[\theta_{t+1} \mid \theta_t] = \theta_t - \eta \cdot \mathbb{E}\left[\sum_{i=1}^M \alpha_i \nabla \hat{R}_i(h_{\theta_t}) \mid \theta_t\right] \tag{10}$$

$$= \theta_t - \eta \cdot \sum_{x,y} \mathbb{E}\left[\sum_{i=1}^M \hat{S}_i(x,y)\alpha_i\right] \nabla L(y, h_{\theta_t}(x)) \tag{11}$$

$$= \theta_t - \eta \cdot \sum_{x,y} T(x,y) \nabla L(y, h_{\theta_t}(x)) \tag{12}$$

$$= \theta_t - \eta \cdot \mathbb{E}\left[\sum_{x,y} \hat{T}(x,y)\right] \nabla L(y, h_{\theta_t}(x)) = \mathbb{E}[\theta_{t+1}^T \mid \theta_t]. \tag{13}$$

$\square$

## C.2 FEDPALS IN THE LIMITS

As $\lambda \to \infty$, because the first term in equation 7 is bounded, the problem shares solution with

$$\min_{\alpha_1,\ldots,\alpha_M} \sum_i \frac{\alpha_i^2}{n_i} \quad \text{s.t.} \quad \sum_i \alpha_i = 1, \quad \forall i : \alpha_i \geq 0. \tag{14}$$

Moreover, we have the following result.

**Proposition 3.** *The optimization problem*

$$\min_\alpha \sum_i \frac{\alpha_i^2}{n_i} \quad s.t \quad \sum_i \alpha_i = 1 \quad \alpha_i \geq 0 \,\forall\, i \,,$$

*has the optimal solution $\alpha_i^* = \frac{n_i}{\sum_i n_i}$ where $i \in [1, m]$*

*Proof.* From the constrained optimization problem we form a Lagrangian formulation

$$\mathcal{L}(\alpha, \mu, \tau) = \sum_i \frac{\alpha_i^2}{n_i} + \mu \underbrace{\left(1 - \sum_i \alpha_i\right)}_{h(\alpha)} + \tau \underbrace{-\alpha}_{g(\alpha)}$$

We then use the KKT-theorem to find the optimal solution to the problem.

$$\nabla_\alpha \mathcal{L}(\alpha^*) = 0 \implies \forall i : \; 2\frac{\alpha_i^*}{n_i} - \mu - \tau = 0. \tag{15}$$

In other words, the following ratio is a constant,

$$\forall i \quad \frac{\alpha_i^*}{n_i} = c$$

for some constant c. We have the additional conditions of primal feasibility, i.e.

$$h(\alpha^*) = 0$$
$$g(\alpha^*) \leq 0$$

From the first constraint, we have $\sum_{i=1}^{M} \alpha_i^* = 1$, and thus,

$$\sum_{i=1}^{M} \alpha_i^* = c \sum_{i=1}^{M} n_i = 1$$

which implies that $c = 1/\sum_{i=1}^{M} n_i$ and thus

$$\forall i : \alpha_i^* = \frac{n_i}{\sum_{i=1}^{M} n_i} \ .$$

$\square$

