# OpenReview forum: "Overcoming label shift in targeted federated learning"
_ICLR.cc/2025/Conference — Submitted to ICLR 2025_

### Official Review · Reviewer_P4Wj · 2024-10-17

**Soundness:** 2
**Presentation:** 2
**Contribution:** 2
**Rating:** 3
**Confidence:** 3

**Summary:**

The paper introduces FedPALS, a method to deal with label shift in federated learning. Label shift happens when label distributions change across clients or between clients and target. The authors propose a novel aggregation scheme to optimize a convex combination of client models to ensure that the aggregated model is better suited for the label distribution of the target domain.

**Strengths:**

S1: The problem of label skews is interesting and well positioned.

S2: The paper gives good theoretical proofs on the algorithm.

**Weaknesses:**

W1: The compared baselines are too old. The authors only compare algorithms before 2020 (FedAvg, FedProx, SCAFFOLD). There are several recent works on label shifts in federated learning (e.g., [1, 2, 3]). Lack of these recent baselines cannot convince that the proposed algorithm is SOTA.

W2: The authors work with C=2,3 and beta=0.1 partitions. How about C=1, the most extreme label shift?

W3: Assumes target label distribution known is often not practical. No examples of how to get this info without privacy risk.

[1] Li, Xin-Chun, and De-Chuan Zhan. "Fedrs: Federated learning with restricted softmax for label distribution non-iid data." Proceedings of the 27th ACM SIGKDD conference on knowledge discovery & data mining. 2021.

[2] Zhang, Jie, et al. "Federated learning with label distribution skew via logits calibration." International Conference on Machine Learning. PMLR, 2022.

[3] Diao, Yiqun, Qinbin Li, and Bingsheng He. "Exploiting Label Skews in Federated Learning with Model Concatenation." Proceedings of the AAAI Conference on Artificial Intelligence. Vol. 38. No. 10. 2024.

**Questions:**

See weaknesses. Given the very weak baselines, it is not convincing that the proposed solution is SOTA to reach the bar of acceptance.

---

> ### Author Response · Authors · 2024-11-21
> **Response to reviewer P4Wj**
>
> We thank P4Wj for taking the time to review our work. We hope that our answers below address any questions and concerns brought up.
>
> - The compared baselines are too old. The authors only compare algorithms before 2020 (FedAvg, FedProx, SCAFFOLD). There are several recent works on label shifts in federated learning (e.g., [1, 2, 3]). Lack of these recent baselines cannot convince that the proposed algorithm is SOTA.
>
>
> A: We are considering the specific setting of label shift between the client aggregate and the target. We also assume the server has access to the label marginals of client and target. This setting is distinct from just label shift between clients and it is unclear whether newer proposed algorithms for the latter, which are not tailored to our setting, would outperform FedProx and SCAFFOLD. Could the reviewer please motivate why the proposed works, which do not handle label shift between client aggregate and target, would be preferable to the baselines we have?
>
> - The authors work with C=2,3 and beta=0.1 partitions. How about C=1, the most extreme label shift?
>
> A: Thank you for the suggestion. However, $C=1$ represents an extreme case where each client only contains a single label, making the problem trivial at the client level. In such scenarios, local classifiers are unlikely to learn meaningful representations due to the lack of label diversity, and the applicability of the learned model to federated aggregation becomes unclear. Nonetheless, we appreciate the insight.
>
> - Assumes target label distribution known is often not practical. No examples of how to get this info without privacy risk.
>
> A: If the target label distribution is unknown the approach we propose is not suitable. It may be the case that we have some other information about the target that could be used, but that is another setting that we do not consider in our work.

---

> > ### Comment · Reviewer_P4Wj · 2024-11-22
> > **Ack**
> >
> > Thank the authors for the reply. However, my concerns are not well addressed. Given other reviews, I keep my score.

---

> > > ### Author Response · Authors · 2024-11-22
> > > **Discussion**
> > >
> > > Thank you for engaging. Seeing as the conference intends for this period to be a discussion, we hope that you are willing to elaborate on why our point-by-point response is not addressing your three concerns:
> > >
> > > W1: Recency does not make a strong baseline. None of the proposed methods address label shift between the target domain and the aggregate of clients. Could you explain why they would make good baselines?
> > > W2: The C=1 setting is pointless since there is no learning signal on the clients when there is a single class.
> > > W3: There is always a privacy-accuracy tradeoff in federated learning. There is a risk even for standard FedAvg. We show there is great gain in accuracy when sharing the label marginal and clients can decide for themselves if it is worth it. This is the premise of the paper and it does not take away from its contributions at all.
> > >
> > > Finally, "Reaching SOTA" is specific to the task at hand. Our task is new (targeted FL with label shift) and there is no SOTA before our paper, not recently and not longer ago. We compare our algorithm to established, general methods for this reason, and show that we perform as well or better than them in every case.

---

### Official Review · Reviewer_otNV · 2024-11-02

**Soundness:** 3
**Presentation:** 3
**Contribution:** 3
**Rating:** 6
**Confidence:** 4

**Summary:**

This paper addresses the problem of label shift in federated learning, where client data distributions differ from a target domain with a known label distribution, but no target samples are available. The authors propose a novel aggregation scheme called FedPALS that optimizes a convex combination of client models to align with the target label distribution, ensuring that the aggregated model minimizes the target risk. The paper provides theoretical justification for the proposed method and demonstrates its effectiveness through extensive empirical evaluation, showing that it outperforms traditional approaches like FedAvg, FedProx, and SCAFFOLD in scenarios with significant distributional shifts.

**Strengths:**

- The proposed FedPALS method is novel and addresses a specific yet significant problem in federated learning, i.e., label shift. The approach of optimizing a convex combination of client models to align with the target label distribution is well-justified.
- The paper is technically sound, with theoretical analysis and extensive empirical evaluation. The experiments cover a variety of scenarios and datasets, demonstrating the robustness and effectiveness of the proposed method.
- The writing is clear and the presentation is well-organized.

**Weaknesses:**

- The paper lacks a reference to the work "Agnostic Federated Learning" by Mohri et al., presented at ICML 2019. This work also addresses federated learning with unknown target distributions and shares some theoretical similarities with the proposed method in this paper. The authors should include this reference in the related work section and provide a more detailed comparison in both the theoretical and experimental sections to highlight the differences and similarities between the two approaches.
- The paper could be improved by providing a more detailed discussion on the limitations of the proposed method. For example, how does FedPALS perform in scenarios where the label shift assumption does not hold? Are there any cases where FedPALS might fail or underperform?
- The paper discusses the choice of the hyperparameter $\lambda$ but could provide more guidance on how to select this parameter in practice. A more detailed analysis of the impact of $\lambda$ on the performance and robustness of the method would be valuable.

**Questions:**

- Could the authors provide a more detailed comparison with existing methods, particularly those that address distributional shifts in federated learning? How does FedPALS differ from and improve upon these methods?
- Does the proposed method generalize to other types of distributional shifts, such as covariate shift? If so, could the authors provide some initial results or discussion on this topic?

---

> ### Author Response · Authors · 2024-11-21
> **Response to reviewer otNV**
>
> We thank otNV for their helpul and insightful review. We hope that our answers below address any questions and concerns brought up.
>
> - The paper lacks a reference to the work "Agnostic Federated Learning" by Mohri et al., presented at ICML 2019. This work also addresses federated learning with unknown target distributions and shares some theoretical similarities with the proposed method in this paper. The authors should include this reference in the related work section and provide a more detailed comparison in both the theoretical and experimental sections to highlight the differences and similarities between the two approaches.
>
> A: We thank the reviewer for pointing this out this omission. We have added AFL as a baseline to our experiments.
>
> - The paper could be improved by providing a more detailed discussion on the limitations of the proposed method. For example, how does FedPALS perform in scenarios where the label shift assumption does not hold? Are there any cases where FedPALS might fail or underperform?
>
> A: We have added some further discussion of the limitations of the method in section 6.
>
> - The paper discusses the choice of the hyperparameter $\lambda$ but could provide more guidance on how to select this parameter in practice. A more detailed analysis of the impact of $\lambda$ on the performance and robustness of the method would be valuable.
>
>  A: We have added further discussion and guidance on the choice of $\lambda$ at the end of section 3.
>
> - Could the authors provide a more detailed comparison with existing methods, particularly those that address distributional shifts in federated learning? How does FedPALS differ from and improve upon these methods?
>
> A: We have some discussion on why we do not use such approaches in the related work section, in particular on lines 348--356.
>
> - Does the proposed method generalize to other types of distributional shifts, such as covariate shift? If so, could the authors provide some initial results or discussion on this topic?
>
> A: Covariate shift is inherently different from label shift as it typically involves changes in the distributions of high-dimensional continuous variables, such as input features. However, in our setting, we assume that $P(Y∣X)$ and $P(X∣Y)$ are invariant across sources and targets, while $P(Y)$ is changing. This change in $P(Y)$ naturally induces a shift in $P(X)$, meaning that covariate shift is inherently present in all of our experiments.
>
> This relationship is evident from the joint distribution $P(X,Y)=P(X∣Y)P(Y)=P(Y∣X)P(X)$. As $P(Y)$ changes, $P(X)$ is also affected, reflecting aspects of covariate shift within our framework. We will add a clarification in the paper to explain this connection and emphasize how our approach implicitly addresses certain cases where covariate shift arises due to changes in label distribution.

---

> > ### Comment · Reviewer_otNV · 2024-11-27
> >
> > Thank you for the response. You have addressed my concerns. Given other reviews, I will keep my score.

---

> ### Author Response · Authors · 2024-11-26
> **Re:**
>
> Dear reviewer otNV,
>
> Thanks again for reviewing our work. We believe our revision and response should address your concerns and kindly ask whether you would consider updating your review. If there are any remaining questions we can answer before the deadline tomorrow, please let us know.
>
> Best, Authors

---

### Official Review · Reviewer_5CXe · 2024-11-03

**Soundness:** 1
**Presentation:** 2
**Contribution:** 1
**Rating:** 3
**Confidence:** 4

**Summary:**

This paper addresses label shift in federated learning, a challenge where label distributions vary across clients and differ from the target domain, potentially leading to degraded model performance. The authors propose FedPALS, a novel model aggregation approach that aligns client updates with the target label distribution, enabling better generalization under label shift. The method incorporates a weighting strategy that balances the need to match target distribution closely while minimizing variance in updates. Experimental results on datasets with label sparsity demonstrate FedPALS’s effectiveness over baseline federated learning methods, showing improved model accuracy in label-shifted target domains.

**Strengths:**

1. FedPALS introduces a well-justified aggregation technique that adjusts model updates for label shifts, making it highly applicable in non-i.i.d. data settings commonly found in real-world federated learning applications.

2. The paper includes extensive experiments on multiple datasets with varying degrees of label sparsity, demonstrating FedPALS’s superiority over standard methods like FedAvg, FedProx, and SCAFFOLD, thus validating its effectiveness in handling label shifts.

**Weaknesses:**

FedPALS lacks novelty, largely building on existing federated aggregation methods without introducing a fundamentally new approach. It uses outdated baselines; incorporating recent models (2023-2024) would provide better comparisons. The experiments with a limited number of clients (e.g., 3 for PACS, 10 for CIFAR-10) restrict evaluation of FedPALS’s scalability in larger federated settings. Additionally, its optimization for client aggregation weights could become computationally expensive with more clients or labels. The method focuses on label shift alone, though real-world federated learning often involves covariate shifts. Lastly, FedPALS relies on a critical hyperparameter, λ, without clear guidance on selection, which may limit practical deployment.

**Questions:**

1. The novelty of the proposed method (FedPALS) is not strongly highlighted, as it builds upon existing model aggregation schemes in federated learning without introducing a fundamentally new approach. Although it adapts the aggregation scheme to account for label shift, similar issues have been addressed with methods like FedAvg and FedProx, making it essential to clarify the unique contributions FedPALS offers beyond these.

2. The baseline used in this paper is too old. Please use some new baselines from 2023 or 2024 for comparison.

3. The experiments are conducted with a limited number of clients (e.g., 3 clients in PACS and 10 in CIFAR-10 and Fashion-MNIST), which is not representative of typical large-scale federated learning environments. This small client network size limits the evaluation of FedPALS's scalability and robustness, especially under diverse, high-client scenarios that federated learning is intended for.

4. FedPALS introduces an optimization process to determine client aggregation weights, which may become computationally expensive as the number of clients or labels increases. The paper does not address the potential scalability issues of this approach, raising concerns about its applicability in large federated learning settings with numerous heterogeneous clients or labels.

5. The method focuses on label shift, assuming that the input distributions remain consistent across clients and the target domain. However, in real-world federated learning, covariate shift often occurs alongside label shift. Expanding FedPALS to handle both types of shifts would enhance its comprehensiveness and applicability.

6. The method’s performance relies on a crucial hyperparameter, λ, which balances bias and variance in the model. The paper does not offer a clear, practical method for choosing λ, which may pose challenges for real-world deployment where fine-tuning may not be feasible. This lack of guidance could hinder the ease of implementation and generalization of FedPALS.

---

> ### Author Response · Authors · 2024-11-21
> **Response to reviewer 5CXe**
>
> We thank 5CXe for their review. We hope that our answers below address any questions and concerns brought up.
>
> - FedPALS lacks novelty, largely building on existing federated aggregation methods without introducing a fundamentally new approach.
>
> A: We respectfully disagree with the reviewer’s assessment that FedPALS lacks novelty. The core contribution of FedPALS lies in the optimization of aggregation weights using both source and target marginal distributions, a strategy that is specifically tailored to address label shifts in federated learning. This is a fundamentally novel approach, as traditional federated aggregation methods, such as FedAvg, FedProx and SCAFFOLD, do not account for mismatches between the client and target label distributions.
>
> - It uses outdated baselines; incorporating recent models (2023-2024) would provide better comparisons.
>
> A: We are considering the specific setting of label shift between the client aggregate and the target where we assume the server has access to the label marginals of client and target. This setting is novel and it is unclear whether newer proposed algorithms which are not tailored to this setting would outperform FedProx and SCAFFOLD. Would the reviewer please advise which more recent models would provide better comparison than the baselines chosen?
>
> - The experiments with a limited number of clients (e.g., 3 for PACS, 10 for CIFAR-10) restrict evaluation of FedPALS’s scalability in larger federated settings.
>
> A: While some of our experiments, such as PACS, use a smaller number of clients, we also evaluate FedPALS in larger federated settings. Specifically, the iWildCam experiment involves 100 clients (see Figure 3b), and for CIFAR-10, we conduct experiments with varying numbers of clients, including up to 100. The CIFAR-10 results are detailed in the appendix. This demonstrates that FedPALS is scalable and effective across a range of client network sizes.
>
> - Additionally, its optimization for client aggregation weights could become computationally expensive with more clients or labels.
>
> A: We respectfully disagree with the reviewer’s concern regarding the computational expense of optimizing client aggregation weights. This optimization is performed only once and is formulated as a quadratic program over $M$ variables, where $M$ is the number of clients. In all our experiments, this computation takes less than a few seconds to complete, making it computationally negligible compared to the significantly more resource-intensive task of training $M$ deep neural networks, which is the primary focus of federated learning.
>
> - The method focuses on label shift alone, though real-world federated learning often involves covariate shifts.
>
> A: We acknowledge the reviewer’s point; however, the focus of this work is specifically on label shift, not covariate shift. While real-world federated learning may involve both types of shifts, addressing label shift is a distinct and challenging problem that warrants dedicated study.
>
> - Lastly, FedPALS relies on a critical hyperparameter, λ, without clear guidance on selection, which may limit practical deployment.
>
> A: We have added further discussion and guidance on the choice of $\lambda$ at the end of section 3.

---

> ### Author Response · Authors · 2024-11-26
> **Re:**
>
> Dear reviewer 5CXe,
>
> Thanks again for reviewing our work. We believe our revision and response should address your concerns and kindly ask whether you would consider updating your review. If there are any remaining questions we can answer before the deadline tomorrow, please let us know.
>
> Best,
> Authors

---

> > ### Comment · Reviewer_5CXe · 2024-11-26
> >
> > Thank you for your reply. Your answer partially addressed my concerns, but I still think your baseline comparison is too simple. Can you compare it with the recent baselines in your scenario?
> >
> > You can refer to the following baselines.
> > [1] FedRC: Tackling Diverse Distribution Shifts Challenge in Federated Learning by Robust Clustering
> > [2] Optimizing the Collaboration Structure in Cross-Silo Federated Learning

---

> ### Author Response · Authors · 2024-11-26
> **Thank you**
>
> Thank you for suggesting specific baselines—this substantially helps the discussion!
>
> We have looked into the two papers you suggested and neither tackle the targeted federated learning scenario—i.e., where the target distribution differs from the client aggregate. Moreover, neither [1] or [2] are directly applicable in our setting:
>
> Both FedRC and FedCollab produce sets of models, one for each cluster or coalition. Neither clustering in [1] or [2] is sensitive to a target distribution, and it is not clear how to pick which cluster's model to apply in the target after training. Without data from the target it can't be clustered with the other clients, so there's no obvious choice for which $\theta_k$ in [1] or $\hat{h}_{\alpha_i}$ in [2] to use in the target. Finally, both [1] and [2] optimize the in-cluster client aggregate error which does not conform to our setting, in which we want to optimize the target distribution error.
>
> Between them, the 5 reviewers have now suggested 6 different baselines, and only 1 baseline (AFL) was suggested by two reviewers. We added AFL to the paper since the revision and, in summary, it does not perform much better than other baselines and still substantially worse than our method. None of the suggested methods are designed for targeted FL.
>
> In light of the above, we hope you agree that our setting is new and that there are no recent or old baselines that are specifically suited to it. Based on this, we firmly argue that our selection of baselines is representative and sufficient. Since this was your primary concern, we hope that you consider updating your review.
>
> We will include the suggested baselines in the related work section and contrast their settings with ours. Let us know if we can clarify anything further.
>
> Best,
> Authors

---

### Official Review · Reviewer_mzUQ · 2024-11-03

**Soundness:** 2
**Presentation:** 3
**Contribution:** 2
**Rating:** 5
**Confidence:** 5

**Summary:**

This paper studied a new FL problem under label shifts where the server knows the different label distributions of clients and target domains are known to the central server but unknown to the clients. To solve this problem, this paper optimized the aggregation weights using the label distributions. Experimental results demonstrated the effectiveness of the proposed methods over several baselines.

**Strengths:**

(S1) A novel FL problem with label shifts is formulated by assuming that the server knows the different label distributions of clients and target domains are known to the central server but unknown to the clients

(S2) A novel parameter aggregation strategy is proposed based on the label distributions. It also balances the effective sample size and the alignment with the target label distribution.

(S3) Experimental results on several data sets demonstrate the effectiveness of the proposed method.

**Weaknesses:**

(1) The training procedures of the proposed FedPALS method based on equation (7) are confusing. The parameter $\lambda$ largely affects the model performance in the experiments. Though section 3.2 provides several options, it is unclear how $\lambda$ is selected in the experiments, e.g., Figure 3(a)(b).

(2) The assumption of this paper is strong.
- It assumes that the target label distribution $T (Y )$ is known to the server, but no training examples are available for the target client. It would be better to provide more realistic FL scenarios to illustrate the importance of this special problem settings.
- It assumes that the server knows the different label distributions of clients. This can also result in the privacy concern.

(3) The prior information regarding $T (Y )$ is only available for the proposed FedPALS approach. It might be unfair for performance comparison between FedPALS and baselines.

**Questions:**

(1) The proof of Proposition 2 is confusing. When $\mu = 2\sum_i \frac{\alpha_i^*}{n_i}$, how can it derive $\mu = \frac{2}{sum_i n_i}$ using the the primal feasibility condition?

(2) How is the parameter $\lambda$ selected in Figure 3(a)(b)?

(3) It seems that the approaches in Figure 3(b) do not converge yet after 80 communication rounds, as their target F1-scores keep increasing.

(4) The paper [c1] also studied the FL problem under unknown target client. It can assume no prior information regarding the target client. Thus, the developed AFL can be one of the strong baselines for FedPALS in the experiments.

[c1] Mohri, Mehryar, Gary Sivek, and Ananda Theertha Suresh. "Agnostic federated learning." In International conference on machine learning, pp. 4615-4625. PMLR, 2019.

---

> ### Author Response · Authors · 2024-11-21
> **Response to reviewer mzUQ**
>
> We thank mzUQ for their thorough and helpful review. We hope that our answers below address your questions and concerns.
>
> - The training procedures of the proposed FedPALS method based on equation (7) are confusing. The parameter
>  largely affects the model performance in the experiments. Though section 3.2 provides several options, it is unclear how
>  is selected in the experiments, e.g., Figure 3(a)(b).
>
> A: The $\lambda$ parameter is selected such that they represent 25%, 50% and 75% of the ESS compared to the FedAvg solution. This is detailed in Appendix A.3 and A.4. We
>
> - The assumption of this paper is strong. It assumes that the target label distribution is known to the server, but no training examples are available for the target client. It would be better to provide more realistic FL scenarios to illustrate the importance of this special problem settings.
>
>
> It assumes that the server knows the different label distributions of clients. This can also result in the privacy concern.
>
> A: We acknowledge that the assumption of the server knowing the target label distribution $T(Y)$ is a strong one. However, this is central to the problem setting we address, where the server can access high-level aggregated label statistics without requiring individual data points from the target domain. This setup is motivated by real-world applications, such as retail or healthcare, where such aggregate statistics are often accessible and privacy-preserving.
>
> Regarding the assumption that the server knows the clients' label distributions, we recognize that this introduces potential privacy concerns. However, as part of the privacy-accuracy tradeoff inherent in federated learning, we believe that the performance gains from utilizing $T(Y)$ and client label information justify exploring this scenario. Notably, sharing aggregate label statistics is less invasive than sharing raw data, making this a reasonable compromise in many practical settings.
>
> - The prior information regarding T(Y) is only available for the proposed FedPALS approach. It might be unfair for performance comparison between FedPALS and baselines.
>
> A: Our conclusions are focused on demonstrating that $T(Y)$ is a valuable signal for federated learning. Even under the assumption of its availability, this does not invalidate the broader significance of leveraging label distribution knowledge for improved performance.
>
> Furthermore, it is unclear how other methods could effectively use $T(Y)$ without substantial modification. To address this, we have combined FedPALS with FedProx in our experiments as it is straightforward to do so, while combining with other baselines like SCAFFOLD is more complex. This ensures a more meaningful comparison of FedPALS’ contribution while recognizing its specialized assumptions.
>
> - The proof of Proposition 2 is confusing.
>
> A: We thank the reviewer for pointing this out. We have changed the proof to be simpler and more straightforward.
>
> - How is the parameter $\lambda$ selected in Figure 3(a)(b)?
>
> A: The $\lambda$ parameter is selected such that they represent 25%, 50% and 75% of the ESS compared to the FedAvg solution. This is detailed in appendix A.3 and A.4 for the respective experiments.
>
> - It seems that the approaches in Figure 3(b) do not converge yet after 80 communication rounds, as their target F1-scores keep increasing.
>
> A: We ackknowledge that the models may not be fully converged here. This experiment is quite large scale and as such takes a lot of compute to perform. We have followed the benchmarking details from Bai et al.
>
> - The paper [c1] also studied the FL problem under unknown target client. It can assume no prior information regarding the target client. Thus, the developed AFL can be one of the strong baselines for FedPALS in the experiments.
> [c1] Mohri, Mehryar, Gary Sivek, and Ananda Theertha Suresh. "Agnostic federated learning." In International conference on machine learning, pp. 4615-4625. PMLR, 2019.
>
> A: We thank the reviewer for pointing out paper c1 which we had not mentioned in our related work. We have added this as a baseline to our tasks, see Table 1 and Figure 3 in the revised manuscript. We are currently running the iWildCam experiment and will update if this gets completed before the deadline.

---

> ### Author Response · Authors · 2024-11-26
> **Re:**
>
> Dear reviewer mzUQ,
>
> Thanks again for reviewing our work. We believe our revision and response should address your concerns and kindly ask whether you would consider updating your review. If there are any remaining questions we can answer before the deadline tomorrow, please let us know.
>
> Best,
> Authors

---

> > ### Comment · Reviewer_mzUQ · 2024-11-26
> > **Comments**
> >
> > Thanks for providing the rebuttal.
> >
> > The proof of Proposition 2 is clear.
> >
> > However, other concerns remain.
> >
> > (1) Though section 3.2 provides heuristic selections for $\lambda$, it is unclear how it is selected in the experiments.
> >
> > (2) The privacy concerns induced by the target label distribution are not solved.
> >
> > (3) The explanation regarding "the models may not be fully converged" is unconvincing. Does it imply that the benchmarking details from Bai et al. do not consider the model convergence? If so, why should such benchmarks be selected? Besides, "this experiment is quite large scale and as such takes a lot of compute to perform". It is still confusing why it does not take a longer time to obtain converged results.

---

> > > ### Author Response · Authors · 2024-11-27
> > > **Response**
> > >
> > > Thank you for your comments. We hope to further clarify the points of concern below.
> > >
> > > 1. In the experiments we choose a set of $\lambda$ values which corresponds to different amounts of ESS. Models are then trained using these lambdas. Then, for the FedPALS-results which we present in Table 1, we select the best performing one of these models using a source validation set.
> > >
> > > 2. The privacy-accuracy tradeoff is not something that we can "solve". We are clear about the setting and what is available in it. For some applications the label marginals may be of such importance that sharing them is unacceptable. Then using the targeted FL setting is not applicable. However, in cases where this is not the case the performance gains we observe in our experiments should warrant consideration. Is there anything specific the reviewer has in mind for us to address regarding this?
> > >
> > > 3. Our comments on computation were referring to the fact that the computation takes a long time to perform, we are training 100 ResNet50 neural nets on a image dataset of >100K images. Performing this experiment takes days.
> > >
> > > We performed the experiments with the hyperparameters which were reported in the previous work. It is unfortunately not feasible in the time-frame available for this discussion period for us to redo the entire experiment to have more communication rounds.

---

### Official Review · Reviewer_VuV9 · 2024-11-08

**Soundness:** 1
**Presentation:** 2
**Contribution:** 3
**Rating:** 3
**Confidence:** 4

**Summary:**

This work addresses the realistic scenario of generating a global model for an arbitrary target distribution that may differ from the distribution of the aggregated client training datasets. To tackle this challenge, the authors assume that the central server has knowledge of the different label distributions of the clients and target domains, while this information remains unknown to the clients. They introduce FedPALS, a method that effectively balances the trade-off between bias (related to the target distribution relative to the aggregated training dataset) and variance stemming from the number of each client dataset.

**Strengths:**

This paper is the first to tackle the realistic federated learning scenario of generating a global model for an arbitrary target distribution that may differ from the aggregated client training datasets.

**Weaknesses:**

**Weaknesses**

The presentation of this paper is notably poor, as it contains numerous inaccuracies and unclear statements throughout. For instance:

1. (L87-88 and L136)  "A common assumption in federated learning is that all client distributions are identical."
-> This is not right. Federated learning generally allows for differences in client distributions, with data heterogeneity being one of the main challenges.

2. (L 97-98)-> the phrase "Given a set of clients S_1, ..., S_M" -> these are the set of client's marginal distribution.

3. (L98)There is a lack of clarity regarding the loss function; the specific loss being used is not mentioned.

4. (L108) "While the server has access to all marginal label distributions," reflects an assumption made in the paper. Given the emphasis on data privacy in federated learning, this assumption requires a much deeper justification. The authors only reference a single paper (Ramakrishna & Dan, 2022) in (L113-115) to support this assumption, which is insufficient. More extensive discussion is needed to validate this critical assumption.

5. (L161) The phrase "As we see in Table 1"  is unclear, as Table 1 appears several pages later. A more helpful reference would be "As we see in Table 1 in Section 5."

6. Proposition 1 discusses the importance of the aggregated training dataset's distribution being equal to that of the target dataset. However, establishing this equivalence alone does not address the challenges of federated learning. The issue of data heterogeneity among clients must be resolved when the aggregated training dataset's distribution matches the target dataset's distribution, particularly when local iterations are performed only once. In practice, due to communication issues, multiple local iterations are typically needed, and the implications of unbiasedness discussed in Proposition 1 are not sufficiently emphasized until lines 285-289.

7. (L 161) There is no definition provided for vector alpha and S  which creates confusion.

8. (L 300) The validation set's specific dataset type is not defined. If it is identical to the target dataset, this would constitute a dangerous approach that could be seen as cheating.

9. The inclusion of the hyperparameter lambda in (L 302) is a significant weakness, as it necessitates searching for optimal lambda values in arbitrary settings, which may vary across different settings.

10. The term "this benchmark" in (L 359) lacks clarity. It should specify what is being referred to. Additionally, the experimental setup appears weak, as modern federated learning research typically uses at least CIFAR-100, with ImageNet or Tiny ImageNet as benchmarks. This study seems to rely on a dataset with fewer than 10 classes, which lacks challenge. Moreover, with only 10 clients and an alpha value of 0.1 for data heterogeneity, it is unclear how performance will change as data heterogeneity increases.

11. In (L 444-445), the phrase "varying the number of clients C across 10 clients" is confusing, as C does not accurately convey the number of clients.

**Questions:**

I am curious about the performance comparison of the proposed method with baseline federated learning algorithms (such as SCAFFOLD) in scenarios where the label distribution of the aggregated training dataset matches that of the target domain.

---

> ### Author Response · Authors · 2024-11-21
> **Response to reviewer VuV9**
>
> We thank VuV9 for their review. As you can see in the revised version of the paper, all of the comments have been addressed.
>
> - (L87-88 and L136) "A common assumption in federated learning is that all client distributions are identical." -> This is not right. Federated learning generally allows for differences in client distributions, with data heterogeneity being one of the main challenges.
>
> A: We have rephrased this paragraph to reflect better our intent: "A common (implicit) assumption in federated learning is that the learned model will be applied in a target domain $T(X,Y)$ that coincides with the marginal distribution of clients ... This is reflected in trained models being evaluated in terms of their average performance over all clients." The identically-distributed-client setting is a special case of that. We are explicitly challenging and extending this assumption to situations with non-identical distributions.
>
> - (L 97-98)-> the phrase "Given a set of clients S_1, ..., S_M" -> these are the set of client's marginal distribution.
>
> A: We have changed the text to clarify that we refer to client distributions.
>
> - (L98)There is a lack of clarity regarding the loss function; the specific loss being used is not mentioned.
>
> A: The risk minimization objective is completely general and not specific to any loss function. In our classification experiments we use the cross-entropy loss, for example. We have added a mention of this in section A of the appendix.
>
> - (L108) "While the server has access to all marginal label distributions," reflects an assumption made in the paper. Given the emphasis on data privacy in federated learning, this assumption requires a much deeper justification. The authors only reference a single paper (Ramakrishna & Dan, 2022) in (L113-115) to support this assumption, which is insufficient. More extensive discussion is needed to validate this critical assumption.
>
> A: The reviewer is correct that there is a privacy concern with sharing label distributions with the server. However, there is always a trade-off between accuracy and privacy in federated learning since model updates also reveal information about clients' data sets. Our work demonstrates a clear benefit in terms of accuracy (in several cases, more than 10\%) if the clients are willing to share their label marginals. It is up to clients to decide whether this tradeoff is worth it for them.
>
> - (L161) The phrase "As we see in Table 1" is unclear, as Table 1 appears several pages later. A more helpful reference would be "As we see in Table 1 in Section 5."
>
> A: We have added this suggestion to the revised manuscript.
>
> - Proposition 1 discusses the importance of the aggregated training dataset's distribution being equal to that of the target dataset. However, establishing this equivalence alone does not address the challenges of federated learning. The issue of data heterogeneity among clients must be resolved when the aggregated training dataset's distribution matches the target dataset's distribution, particularly when local iterations are performed only once. In practice, due to communication issues, multiple local iterations are typically needed, and the implications of unbiasedness discussed in Proposition 1 are not sufficiently emphasized until lines 285-289.
>
> A: It is correct that Proposition 1 does not give guarantees for the multiple-local-iterations case, and we never claim that it does. As the reviewer points out, we already made this remark on ln 285-289. We have moved this remark closer to the proposition itself. Nevertheless, our result is justified in the SGD setting under known label shift, strictly more general than the justification for FedAvg, and we demonstrate empirically that there is a strong benefit to our aggregation scheme also in the non-SGD setting.
>
> - (L 161) There is no definition provided for vector alpha and S which creates confusion.
>
> A: L 161 does not contain a reference to either $\alpha$ or $S$ so it is difficult to know which version of $\alpha$ is intended. Generally, $\alpha$ refers to the weights of a convex combination. It is used for the first time in the definition of the simplex before equation (3). It is then defined for specific combinations in (3), (4), (6), and (7). $S(Y)$ refers to the matrix of all client label marginals. We have clarified this in ``Case 2'' on page 6.

---

> > ### Author Response · Authors · 2024-11-21
> > **Response cont.**
> >
> > - (L 300) The validation set's specific dataset type is not defined. If it is identical to the target dataset, this would constitute a dangerous approach that could be seen as cheating.
> >
> > A: The validation set is an in-domain validation set. The data in this set is drawn from the same distribution as the training data. We do not validate on the target set, but in some experiments we modify the validation set to look more closely like the target e.g. inducing the same label sparsity as in the target (PACS experiment).
> >
> > - The inclusion of the hyperparameter lambda in (L 302) is a significant weakness, as it necessitates searching for optimal lambda values in arbitrary settings, which may vary across different settings.
> >
> > A: We agree that the use of a hyperparameter introduces the need to find a good value for it. However, the introduction of a hyperparameter is hardly a significant weakness. This is done in many proposed methods such as the FedProx baseline [1], but also other methods such as FedDG [2] and FedSR [3] just to name a few.
> >
> > To mitigate the challenge of selecting $\lambda$, we propose leveraging access to the target label distribution (if available) to construct a local validation set by upsampling or downsampling client data to better approximate the target distribution. This approach allows for a more principled and efficient selection of $\lambda$, reducing the need for exhaustive hyperparameter searches. We will clarify this strategy in the revised manuscript.
> >
> > Additionally, our experiments show that FedPALS is robust to a range of $\lambda$ values, suggesting that exact tuning may not always be necessary. For example, a heuristic selection based on effective sample size can often provide satisfactory performance without extensive validation.
> >
> > [1] Tian Li, Anit Kumar Sahu, Manzil Zaheer, Maziar Sanjabi, Ameet Talwalkar, and Virginia Smith.
> > Federated optimization in heterogeneous networks. Proceedings of Machine learning and systems,
> > 2:429–450, 2020.
> > [2] Quande Liu, Cheng Chen, Jing Qin, Qi Dou, and Pheng-Ann Heng. Feddg: Federated domain
> > generalization on medical image segmentation via episodic learning in continuous frequency space.
> > In Proceedings of the IEEE/CVF Conference on Computer Vision and Pattern Recognition, pages
> > 1013–1023, 2021.
> > [3] A Tuan Nguyen, Ser-Nam Lim, and Philip HS Torr. Fedsr: Simple and effective domain generalization
> > method for federated learning. International Conference on Learning Representations, 2022.
> >
> > - The term "this benchmark" in (L 359) lacks clarity. It should specify what is being referred to. Additionally, the experimental setup appears weak, as modern federated learning research typically uses at least CIFAR-100, with ImageNet or Tiny ImageNet as benchmarks. This study seems to rely on a dataset with fewer than 10 classes, which lacks challenge. Moreover, with only 10 clients and an alpha value of 0.1 for data heterogeneity, it is unclear how performance will change as data heterogeneity increases.
> >
> > A: The benchmark is the one by Bai et al. which we refer to in the previous sentence: ``Bai et al (2024) introduced a benchmark ...''. We have clarified this reference in the revised paper. Regarding the experimental setup, we respectfully disagree with the reviewer’s assessment. For scaling to many clients, we have the iWildCam dataset task which we test with 100 clients. Furthermore, the reviewer's suggested benchmark tasks do not have clear label shift and would not be useful to test our approach. Additionally, we conduct experiments on CIFAR-10 with varying numbers of clients, which are provided in the appendix, to further illustrate the robustness of FedPALS under different client configurations.
> >
> > - In (L 444-445), the phrase "varying the number of clients C across 10 clients" is confusing, as C does not accurately convey the number of clients.
> >
> > A: Indeed this should state 'labels per client'. We have fixed this typo in the updated manuscript.
> >
> > - I am curious about the performance comparison of the proposed method with baseline federated learning algorithms (such as SCAFFOLD) in scenarios where the label distribution of the aggregated training dataset matches that of the target domain.
> >
> > A: The method we propose is equivalent to FedAvg in this setting since the FedAvg aggregation scheme minimizes our combination-selection objective when the target and aggregate client distributions match. We make this point in Case 1 in Section 3.2.

---

> > > ### Comment · Reviewer_VuV9 · 2024-11-23
> > >
> > > **Response to Authors:**
> > >
> > > Thank you for your response. Upon reviewing the revised version, I still believe there are significant issues with the overall presentation, which need to be addressed more carefully. Specifically, the paper continues to suffer from unclear terminology, typographical errors, and a lack of sufficient clarification in key areas. These issues were present in the initial version and have not been fully resolved in the current version.
> > >
> > > For example, the setting described in the paper, such as having only one local iteration, does not align with typical federated learning scenarios and occupies an unnecessarily large portion of the discussion. This misalignment could lead to confusion and distract from the overall contribution of the paper.
> > >
> > > One particular point of concern is the use of performance as a defense for privacy issues, which I do not find to be a valid argument. This approach does not convincingly address the core privacy concerns in federated learning.
> > >
> > > Additionally, the statement, "The method we propose is equivalent to FedAvg in this setting since the FedAvg aggregation scheme minimizes our combination-selection objective when the target and aggregate client distributions match," is unclear. I am unsure whether this holds true in the context provided, and further explanation is needed to clarify this assertion.
> > >
> > > After considering the other reviews, I will keep my score as is.

---

> > > > ### Author Response · Authors · 2024-11-25
> > > > **Response**
> > > >
> > > > Thank you for responding to our rebuttal. Any feedback is helpful for improving our paper, so we hope to clear up a few more things.
> > > >
> > > > * Only *Proposition 1* is limited to a single update, but our *algorithm* can be applied on any model update, no matter how it was computed, just like e.g., federated averaging. This is clear from our remark to Proposition 1. All of our experiments run a full epoch on each client before aggregating and we investigate how results vary with epochs in appendix Figure 8. We could certainly reduce the space given to Proposition 1, but we believe that it is important to understand cases when our algorithm is guaranteed to be successful, even if many papers ignores this question.
> > > > * When it comes to privacy, we hope you agree that other works aimed at improving generalization in federated learning also cannot precisely characterize their privacy risk from sharing gradients or model updates. In our case, we can be very precise about what additional information is revealed, and the clients can each choose whether that is worth it for them.
> > > > * For the equivalence with FedAvg in the case where the client distribution aggregate equals the target distribution, this is proven formally in Proposition 3 in the appendix. The minimizer of the regularization term *is* the FedAvg weights. The FedAvg weights are well known to match the client aggregate distribution. Thus, when the target equals the client aggregate, the FedAvg weights minimize our objective.
> > > >
> > > > Sincerely, the authors

---

### Author Response · Authors · 2024-11-21
**General response**

We thank the reviewers for their time and effort in evaluating our work and for their helpful comments which we have used to improve our paper. We have responded to each reviewer and are confident that our rebuttal responds to the concerns which were brought up. Below we provide a summarised list of main points of our rebuttal and changes we have made to the revised manuscript. In the manuscript, changes are marked with blue text for your convenience.

* Added Baseline: hybrid FedProx+FedPals  (Figures 2c, 3a,b)
* Added Baseline: Agnostic Federated Learning (AFL), suggested by Reviewers mzUQ, otNV, to tasks other than iWildCam, (Table 1, Figures 2c-3a)
* Added discussion regarding the label distributions being known by the server (Reviewer VuV9, mzUQ)
* Added discussion on covariate shift (Reviewer )5CXe, otNV)
* Added discussion on the privacy-accuracy tradeoff involved in clients sharing label distribution (Reviewer mzUQ, P4Wj)
* Extended guidelines for the choice of hyperparameter $\lambda$ (Reviewer VuV9, mzUQ, 5CXe)
* Clarified proof of Proposition 2 (Reviewer mzUQ)
* Incorporated all proposed changes and clarifications to the text by Reviewer VuV9
* MOVED the synthetic experiment to the appendix to accommodate these changes.
* Explained why the computational cost of computing the weighting $\alpha$ is insignificant compared to training the model

---

> ### Author Response · Authors · 2024-11-28
> **AFL on iWildCam**
>
> We just wanted to update the reviewers to the fact that the iWildCam experiment has finished and the AFL baseline does not appear to learn effectively at all, only reaching ~0.005 F1 on average. This failure is in line with the results reported in Bai et al. where they report a large reduction in performance in the heterogeneous setting for AFL. See the results for $\lambda=0$ in Table 3 in their paper for reference.

---

### Meta-Review · Area_Chair_MJnQ · 2024-12-19

**Metareview:**

**Summary:** The paper proposes a novel method to address label shifts in federated learning by optimizing model aggregation weights based on known target label distributions. The authors provide theoretical justifications for their approach and conduct empirical evaluations across several datasets, demonstrating improvements over standard baselines like FedAvg and FedProx. The proposed method aims to balance bias and variance while aligning the aggregated model with the target distribution, addressing challenges associated with label shifts in non-i.i.d. data settings.

**Decision:** Despite addressing an important problem and proposing a reasonable approach, the paper has significant shortcomings that limit its potential impact. Several reviewers raised concerns about the method's assumptions, including the requirement for knowledge of the target label distribution and client label marginals, which is often impractical in real-world scenarios. The experimental setup was also criticized for relying on outdated baselines and limited scalability evaluations. More importantly, the presentation of this paper could also be further improved. Because of this, the novelty of the novelty of this paper is challenging to recognize; problem setups, terminology definition, and some experimental details (hyper-parameter tuning) are unclear and confusing. While the authors addressed some concerns during the rebuttal, such as adding new baselines and clarifying theoretical contributions, the responses did not fully alleviate reviewers' critiques.

These concerns outweigh the contributions, leading to the decision to reject. During the reviewer-AC discussion period, no objections were raised to this decision.

**Additional Comments On Reviewer Discussion:**

During the discussion period, the reviewers extensively debated the paper's merits and weaknesses. The main points raised included the practicality of assumptions (e.g., access to target label distributions), the novelty of the approach compared to existing methods, and the scalability of experiments. The authors attempted to address these concerns in their rebuttal by adding new baselines (e.g., AFL) and expanding discussions on privacy-accuracy trade-offs and theoretical justifications. However, reviewers found the additional baselines insufficiently representative of recent advancements in federated learning, and the theoretical explanations did not adequately address concerns about the practicality of assumptions or the computational feasibility of the proposed method. As the AC, I weighed these discussions and concluded that the unresolved concerns about the novelty, scalability, and practicality outweigh the contributions, leading to the decision to reject this paper.

---

### Decision · Program_Chairs · 2025-01-22

Reject